# *Con4m*: UNLEASHING THE POWER OF CONSISTENCY AND CONTEXT IN CLASSIFICATION FOR BLURRED-SEGMENTED TIME SERIES

## ABSTRACT

Blurred-Segmented Time Series (BST) has emerged as a prevalent form of time series data in various practical applications, presenting unique challenges for the Time Series Classification (TSC) task. The BST data is segmented into continuous states with inherently blurred transitions. These transitions lead to inconsistency in annotations among different individuals due to experiential differences, thereby hampering model training and validation. However, existing TSC methods often fail to recognize label inconsistency and contextual dependencies between consecutive classified samples. In this work, we first theoretically clarify the connotation of valuable contextual information. Based on these insights, we incorporate prior knowledge of BST data at both the data and class levels into our model design to capture effective contextual information. Furthermore, we propose a label consistency training framework to harmonize inconsistent labels. Extensive experiments on two public and one private BST data fully validate the effectiveness of our proposed approach, *Con4m*, in handling the TSC task on BST data.

## 1 INTRODUCTION

Time series classification (TSC) has been widely studied in the field of machine learning for many years (Middlehurst et al., 2023). With the rapid development of measurement technology recently, TSC has been extended to various applications in diverse practical domains, such as healthcare (Rafiei et al., 2022; Chen et al., 2022), finance (Dezhkam et al., 2022; Liu & Cheng, 2023), and environmental monitoring (Yuan et al., 2022; Tian et al., 2023). TSC often involves in classifying time series samples into predefined categories with labels and is usually based on the assumption of independence and identical distribution (*i.i.d.*) (Dempster et al., 2021; Zhao et al., 2023).

In practical applications, however, a large number of **Blurred-Segmented Time Series (BST)** data have emerged, which differ in fundamental ways from traditional TSC data: **(1) BST intrinsically records blurred transitions on the boundaries between different states.** For example, in terms of a person's emotional state, the transition from sadness to happiness is ambiguous, with no clear boundaries. **(2) States last for a long duration, segmenting BST.** Take sleep data covering physiological signals of subjects overnight as an example, it shows alternations of different sleep stages, each of which stably lasts for a prolonged period.

The characteristics of BST pose new challenges for mainstream TSC models. **Firstly, the presence of blurred boundaries leads to inconsistent annotations.** In the case of raw BST data, manual annotations usually determine the start and end points of a particular state. Especially in the healthcare domain, data is collected from different hospitals. Due to the lack of standardized quantification criteria, annotations from different doctors vary for their individual experiences. In the TSC task, each type of states is assigned a unique label. Therefore, the inconsistency in labeling across different data sources hampers model training. However, most existing TSC works model time series data by assuming noise-free labels, which significantly limits their performance on BST data.

**The continuous states and gradual transitions call for more coherent contextual prediction.** In the TSC task, BST data is divided into time segments corresponding to different states (labels) to be classified. There are natural temporal dependencies between consecutive segments, which not only exists at the data level but also manifests in the changes of labels. However, mainstream TSC

models (Middlehurst et al., 2023; Foumani et al., 2023) are often designed for publicly available datasets (Bagnall et al., 2018; Dau et al., 2019) based on *i.i.d.* samples, disregarding the inherent contextual dependencies between the samples in time series data. Although some time series models (Shao et al., 2022; Nie et al., 2023) take contextual information of the input data into consideration for predictions with patch-by-patch modeling, they fail to incorporate the class information of consecutive classified time segments so as to achieve coherent predictions for BST data.

To better model BST data, we first analyze how to enhance the relevance between input data and labels in classification tasks by introducing effective contextual information from an information-theoretic perspective. Subsequently, based on the theoretic insights, we incorporate contextual prior knowledge of BST data from both the data and label perspectives to improve the prediction ability of the model. Lastly, drawing inspiration from noisy label learning, we enable the model to progressively harmonize inconsistent labels during the learning process of classification. Consequently, we propose *Con4m* (pronounced **Conform**) - a label **Con**sistency training framework with effective **Con**textual information, achieving **Co**herent predictions and **Con**tinuous representations for ti**m**e series classification on BST data. Extensive experiments on two public and one private BST data demonstrate the superior performance of *Con4m*. In addition, we verify the *Con4m*'s ability to harmonize inconsistent labels by the label substitution experiment. A case study is also shown to give further insight into how *Con4m* works well for BST data.

Our contributions are as follows. **(1)** We are the first to emphasize the importance of BST data and systematically analyze and model it, which is critical for various practical applications. **(2)** We theoretically elucidate the valuable contextual information for the input data in the classification task. Combined with the theoretical insights, we propose a novel framework *Con4m* that can be effectively applied to the TSC task with BST data. **(3)** Extensive experiments fully highlight the superiority of *Con4m* for modeling BST data, shedding light on the era of personalized services when applications like precision medicine, physiological status monitoring and others will prevail.

## 2 VALUABLE CONTEXTS ENHANCE PREDICTIVE ABILITY

Intuitively, it is widely believed that the performance of models on the classification task can be enhanced by incorporating contextual information. But why does this conclusion hold? What kind of contextual information should be introduced? In this section, we aim to analyze this phenomenon from an information-theoretic perspective at a macro level.

Assuming that the random variables of the classified samples and their corresponding labels are denoted as $x_t$ and $y_t$, respectively. $\mathbb{A}_t$ represents the contextual sample set introduced for $x_t$. $x_{\mathbb{A}_t}$ denotes the random variable for the contextual sample set.

**Proposition 1.** *Introducing contextual information does not compromise the performance of a model for the classification task.*

*Proof.*
$$\mathbb{I}(y_t; x_t, x_{\mathbb{A}_t}) = \mathbb{I}(y_t; x_{\mathbb{A}_t}|x_t) + \mathbb{I}(y_t; x_t) \geq \mathbb{I}(y_t; x_t). \tag{1}$$
The inequality holds due to the non-negativity of conditional mutual information. Mutual information measures the correlation between two variables. In the classification task, a higher correlation between samples and labels indicates that the samples are more easily distinguishable by the labels. Based on the assumption that a model can perfectly capture these correlations, a higher mutual information implies a higher upper bound on the model's performance in classifying samples. □

According to (1), the increase in $I(y_t; x_{A_t}|x_t)$ determines the extent to which the upper bound of the model's performance improves. Hence, we employ Theorem 1 to elucidate the specific contextual sample set that can maximize the information gain $I(y_t; x_{A_t}|x_t)$.

**Theorem 1.** *Introducing a contextual sample set that maximizes the predictive ability of labels yields the maximum information gain.*

*Proof.* Expanding $\mathbb{I}(y_t; x_{\mathbb{A}_t}|x_t)$, we have:
$$\mathbb{I}(y_t; x_{\mathbb{A}_t}|x_t) = \sum_{x_t} p(x_t) \sum_{x_{\mathbb{A}_t}} \sum_{y_t} p(y_t, x_{\mathbb{A}_t}|x_t) \log \frac{p(y_t, x_{\mathbb{A}_t}|x_t)}{p(y_t|x_t)p(x_{\mathbb{A}_t}|x_t)} \tag{2}$$

$$= \sum_{\mathrm{x}_t} p(\mathrm{x}_t) \sum_{\mathrm{x}_{\mathbb{A}_t}} \sum_{\mathrm{y}_t} p(\mathrm{y}_t|\mathrm{x}_t, \mathrm{x}_{\mathbb{A}_t}) p(\mathrm{x}_{\mathbb{A}_t}|\mathrm{x}_t) \log \frac{p(\mathrm{y}_t|\mathrm{x}_t, \mathrm{x}_{\mathbb{A}_t})}{p(\mathrm{y}_t|\mathrm{x}_t)} \tag{3}$$

$$= \sum_{\mathrm{x}_t} p(\mathrm{x}_t) \sum_{\mathrm{x}_{\mathbb{A}_t}} p(\mathrm{x}_{\mathbb{A}_t}|\mathrm{x}_t) D_{\mathrm{KL}}(p(\mathrm{y}_t|\mathrm{x}_t, \mathrm{x}_{\mathbb{A}_t}) \| p(\mathrm{y}_t|\mathrm{x}_t)). \tag{4}$$

Given a fixed classification sample $\mathrm{x}_t$ and the inherent distribution $p(\mathrm{y}_t|\mathrm{x}_t)$ of the data, the KL divergence is a convex function that attains its minimum at $p(\mathrm{y}_t|\mathrm{x}_t, \mathrm{x}_{\mathbb{A}_t}) = p(\mathrm{y}_t|\mathrm{x}_t)$. As $p(\mathrm{y}_t|\mathrm{x}_t, \mathrm{x}_{\mathbb{A}_t})$ approaches the boundary of the probability space, indicating a stronger predictive ability for $\mathrm{y}_t$, the value of KL divergence increases. Due to the convexity of KL divergence, there exists a contextual sample set in the data that maximizes $D_{\mathrm{KL}}(p(\mathrm{y}_t|\mathrm{x}_t, \mathrm{x}_{\mathbb{A}_t}) \| p(\mathrm{y}_t|\mathrm{x}_t))$. We denote this sample set as $\mathbb{A}_t^*$ and the maximum KL divergence value as $D_t^*$. Additionally, we note that $\sum_{\mathrm{x}_{\mathbb{A}_t}} p(\mathrm{x}_{\mathbb{A}_t}|\mathrm{x}_t) = 1$. Hence, we can obtain the upper bound for the information gain $\mathbb{I}(\mathrm{y}_t; \mathrm{x}_{\mathbb{A}_t}|\mathrm{x}_t) \leq \sum_{\mathrm{x}_t} p(\mathrm{x}_t) \sum_{\mathrm{x}_{\mathbb{A}_t}} p(\mathrm{x}_{\mathbb{A}_t}|\mathrm{x}_t) D_t^* \leq \sum_{\mathrm{x}_t} p(\mathrm{x}_t) D_t^*$.

To achieve this upper bound, the model needs to introduce a contextual sample set $\mathbb{A}_t^*$ for each sample that maximally enhances its label's predictive ability. Moreover, the model needs to reach an optimal selection strategy distribution $p(\mathrm{x}_{\mathbb{A}_t^*}|\mathrm{x}_t) = 1, p(\mathrm{x}_{\mathbb{A}_t}|\mathrm{x}_t) = 0$ (for $\mathbb{A}_t \neq \mathbb{A}_t^*$). $\qquad \square$

According to Theorem 1, the model needs to find the optimal contextual sample set that enhances its predictive ability. In this paper, we utilize learnable weights to allow the model to adaptively select potential contextual samples. Through explicit supervised learning, the model can directly enhance its predictive ability in an end-to-end manner. On the other hand, benefiting from an information-theoretic perspective, $\mathrm{x}_{\mathbb{A}_t}$ not only includes the raw data of contextual samples but also incorporates their label information, which can be represented as $\mathrm{y}_{\mathbb{A}_t}$. Therefore, we can introduce contextual information at both the data and class levels to enhance the model's predictive ability.

## 3 THE *Con4m* METHOD

In this section, we introduce the details of *Con4m*. Based on the insights of Theorem 1, we introduce effective contextual information at both the data (Sec. 3.1) and class (Sec. 3.2) levels to enhance the predictive ability of *Con4m*. In Sec. 3.3, inspired by the idea of noisy label learning, we propose a label harmonization framework to achieve label consistency. Before delving into the details of *Con4m*, we first provide the formal definition of the time series classification task in our work.

**Definition 1.** *Given a time interval comprising of $T$ consecutive time points, denoted as $s = \{s_1, s_2, \ldots, s_T\}$, a $w$-length sliding window with stride length $r$ is employed for segmentation. $s$ is partitioned into $L$ time segments, represented as $x = \{x_i = \{s_{(i-1) \times r+1}, \ldots, s_{(i-1) \times r+w}\} | i = 1, \ldots, L\}$. The model is tasked with predicting labels for each time segment (sample) $x_i$.*

### 3.1 CONTINUOUS CONTEXTUAL REPRESENTATION ENCODER

BST data exhibits temporal persistence for each class. By paying closer attention to and aggregating neighboring segments, the model can acquire temporally smoother representations of time segments. Smoother representations lead to smoother predictive probabilities. This benefits not only the prediction of consecutive time segments belonging to the same class with the same label but also aligns with the gradual nature of class transitions. Therefore, we introduce the Gaussian prior to allow for a more targeted selection of the contextual sample set $\mathbb{A}_t$ to enhance the model's predictive ability.

Self-attention in BERT (Devlin et al., 2019) has the ability to globally model sequences. However, point-wise attention computations often fail to obtain smooth representations after aggregation. Therefore, similar to the Gaussian filter technique, we use the Gaussian kernel $\Phi(x, y|\sigma)$ as prior weights to aggregate neighbors to obtain smoother representations. Since the neighbors of boundary segments may belong to different classes, we allow each segment to learn its own scale parameter $\sigma$. Formally, as Figure 1 shows, the two-branch **Con-Attention** in the $l$-th layer is:

$$Q, K, V_t, \sigma, V_s = c^{l-1} W_Q^l, c^{l-1} W_k^l, c^{l-1} W_{V_t}^l, c^{l-1} W_\sigma^l, c^{l-1} W_{V_s}^l, \tag{5}$$

$$T^l = \mathrm{SoftMax}\left(\frac{QK^T}{\sqrt{d}}\right), \tag{6}$$

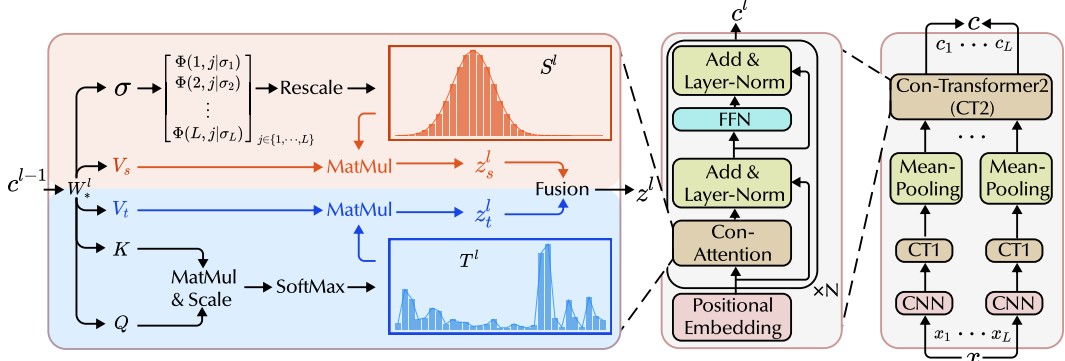

Figure 1: Overview of the encoder of *Con4m*. The leftmost part shows the details of Con-Attention. The right part of the figure shows the architecture of Con-Transformer and the whole encoder of *Con4m*.

$$S^l = \text{Rescale}\left(\left[\frac{1}{\sqrt{2\pi}\sigma_i}\exp\left(-\frac{|j-i|^2}{2\sigma_i^2}\right)\right]_{i,j\in\{1,...,L\}}\right), \tag{7}$$

$$z_t^l = T^l V_t, \quad z_s^l = S^l V_s, \tag{8}$$

where $L$ is the number of the consecutive segments, $d$ is the dimension of the hidden representations, $c^{l-1} \in \mathbb{R}^{L\times d}$ is the $l-1$-th layer's hidden representations and $W_*^l \in \mathbb{R}^{d\times d}$ are all learnable matrices. Rescale($\cdot$) refers to row normalization by index $i$. $Q$, $K$ and $V$ vectors represent the query, key and value of the self-attention mechanism respectively. To distinguish between two computational branches, we use $s/S$ to represent the branch based on Gaussian prior, and $t/T$ to represent the branch based on vanilla self-attention. $T^l$ and $S^l$ are the aggregation weights of the two branches.

Then we use the conventional attention mechanism (Bahdanau et al., 2015) to adaptively fuse $z_t^l$ and $z_s^l$. Finally, as illustrated in Figure 1, by stacking the multi-head version of Con-Attention layers, we construct Con-Transformer, which serves as the backbone of the continuous encoder of *Con4m* to obtain final representations $c$. During the practical implementation, we adopt the same approach proposed by Xu et al. (2022) for the computation of Gaussian kernel function.

## 3.2 CONTEXT-AWARE COHERENT CLASS PREDICTION

In the classification task of BST data, consecutive time segments not only provide context at the data level but also possess their own class information. For instance, in the case of human motion recognition, if an individual is walking at the beginning and end within a reasonable time range, it is highly likely that the intermediate states also corresponds to walking. Existing TSC models (Middlehurst et al., 2023; Foumani et al., 2023) primarily focus on classifying independent segments, overlooking the temporal dependencies of the labels. But our theoretic framework allows for the incorporation of contextual information at the class level into the model's design.

**Neighbor Class Consistency Discrimination.** According to Theorem 1, we aim to identify a set of contextual samples that maximizes the model's predictive ability at the class level. Since directly optimizing the label aggregation is challenging, we adopt the approach of aggregating predictions of segments belonging to the same class. The idea is inspired by the observation that for graph neural networks based on the homophily assumption, aggregating neighbor information belonging to the same class can improve predictive performance (McPherson et al., 2001; Zhu et al., 2020). Therefore, we train a discriminator to determine whether two segments belong to the same class. The model then selects a contextual sample set based on the discriminator's predictions. As the left part of Figure 2 shows, we formalize this process as the following equations:

$$\hat{p} = \text{SoftMax}\left(\text{MLP}_1\left(c\right)\right), \quad Q, K, V = c, c, \hat{p}, \tag{9}$$

$$\hat{R} = \text{SoftMax}\left(\left[\text{MLP}_2\left(Q_i \| K_j\right)\right]_{i,j\in\{1,...,L\}}\right), \tag{10}$$

$$\tilde{p} = \hat{R}_{:,:,1}V, \tag{11}$$

where $\hat{R} \in \mathbb{R}^{L \times L \times 2}$ is the probability of whether two neighbor segments belong to the same class and $(\cdot\|\cdot)$ denotes tensor concatenation. We define the two losses for the model training as:

$$\ell_1 = \text{CrossEntropy}(\hat{p}, y), \quad \ell_2 = \text{CrossEntropy}(\hat{R}, \tilde{Y}), \tag{12}$$

where $\tilde{Y} = [\mathbf{1}_{y_i=y_j}]_{i,j \in \{1,\ldots,L\}}$. Given that $\ell_1$ and $\ell_2$ are of the same magnitude, we equally sum them as the final loss.

**Prediction Behavior Constraint.** Although we incorporate the contextual class information, we still cannot guarantee the overall predictive behavior of consecutive segments. For the BST data, within a suitably chosen time interval, the majority of consecutive time segments span at most two classes. Therefore, the predictions in the intervals should exhibit a constrained monotonicity.

As shown in Figure 2, for each class in prediction results, there are only four prediction behaviors for consecutive time segments, namely *high confidence, low confidence, confidence decreasing, and confidence increasing*. To constrain the behavior, we use function fitting to integrate $\tilde{p}$. Considering the wide applicability, we opt for the hyperbolic tangent function (*i.e.*, Tanh) as our basis. Formally, we introduce four tunable parameters to exactly fit the monotonicity as:

$$\bar{p} = \text{Tanh}(x|a, k, b, h) = a \times \text{Tanh}(k \times (x + b)) + h, \tag{13}$$

where parameter $a$ constrains the range of the function's values, $k$ controls the slope of the transition of the function, $b$ and $h$ adjust the symmetry center of the function, and $x$ is the given free vector in the x-coordinate. We use the MSE loss to fit the contextual predictions $\tilde{p}$ as follows:

$$\ell_3 = \|\text{Tanh}(x|a, k, b, h) - \tilde{p}\|^2. \tag{14}$$

It deserves to emphasize that $\tilde{p}$ in this fit has no gradient and therefore does not affect the parameters of the encoder. Please see Appendix B for more fitting details.

After function fitting, we obtain independent predictions $\hat{p}$ for each segment and constrained predictions $\bar{p}$ that leverage the contextual class information. For the inference stage, we use the average of them as the final coherent predictions, *i.e.*, $\hat{y} = \arg\max(\hat{p} + \bar{p})/2$. Next, we demonstrate how these predictions are combined during the training phase to achieve harmonized labels.

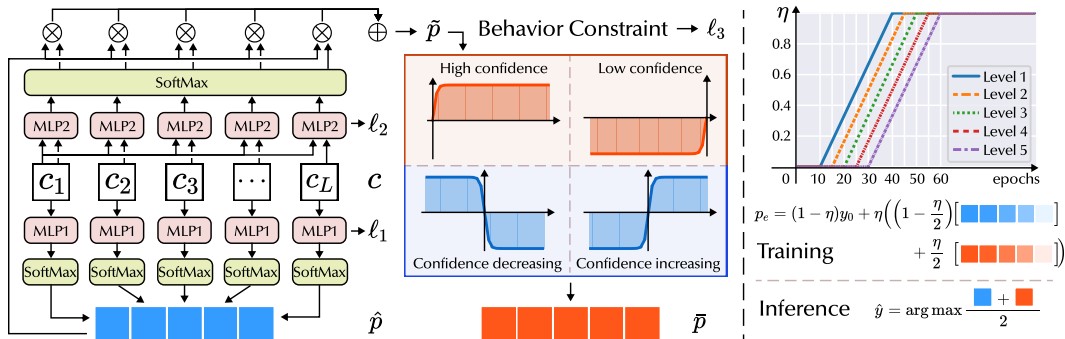

Figure 2: Overview of context-aware coherent class prediction and consistent label training framework in *Con4m*. The left part describes the neighbor class consistency discrimination task and the prediction behavior constraint. The rightmost part presents the training and inference details for label harmonization.

## 3.3 CONSISTENT LABEL TRAINING FRAMEWORK

Due to inherent blurred boundary, the annotation of BST data often lacks quantitative criteria, resulting in experiential differences among individuals. Such discrepancies are detrimental to models and we propose a training framework to enable the model to adaptively harmonize inconsistent labels.

**Learning from easy to hard.** We are based on a fact that although people may have differences for the blurred transitions between states, they tend to reach an agreement on the most significant core part of the states. In other words, the empirical differences become more apparent when approaching the transitions. Therefore, we adopt curriculum learning techniques to help the model learn samples from the easy (core) to the hard (transition) part. Formally (see the diagram in Figure 6(a) in

Appendix), for a continuous $K$-length state, we divide it into $N_l = 5$ equally sized levels as follows:

$$\left( \lceil (N_l - 1) \frac{K}{2N_l} \rceil, \lfloor (N_l + 1) \frac{K}{2N_l} \rfloor \right) ; \cdots ; \left[ 1, \lceil \frac{K}{2N_l} \rceil \right) \bigcup \left( \lfloor (2N_l - 1) \frac{K}{2N_l} \rfloor, K \right]. \quad (15)$$

Then we sample the same number of time intervals from each level. The higher the level, the more apparent the inconsistency. Therefore, as Figure 2 shows, during the training stage, *Con4m* learns the corresponding intervals in order from low to high levels, with a lag gap of $E_g = 5$ epochs.

**Harmonizing inconsistent labels.** Inspired by the idea of noisy label learning, we gradually change the original labels to harmonize the inconsistency. The model preferentially changes the labels of the core segments that are easier to reach a consensus, which can avoid overfitting of uncertain labels. Moreover, the model will consider both the independent and contextual predictions to robustly change inconsistent labels. Specifically, given the initial label $y_0$, we update the labels $y_e = \arg\max p_e$ for the $e$-th epoch, where $p_e$ is obtained as follows:

$$\omega(e, 5) = \text{Rescale}\left( [\exp((e - m)/2)]_{m \in \{0, \ldots, 4\}} \right), \quad (16)$$

$$\hat{p}_e^5 = \omega(e, 5) \cdot [\hat{p}_{e-m}]_{m \in \{0, \ldots, 4\}}, \quad \bar{p}_e^5 = \omega(e, 5) \cdot [\bar{p}_{e-m}]_{m \in \{0, \ldots, 4\}}, \quad (17)$$

$$p_e = (1 - \eta) y_0 + \eta \left( \left(1 - \frac{\eta}{2}\right) \hat{p}_e^5 + \frac{\eta}{2} \bar{p}_e^5 \right), \quad (18)$$

where $\hat{p}_{e-m}$ and $\bar{p}_{e-m}$ are the independent and contextual predictions in the $e - m$-th epoch respectively and $\cdot$ denotes the dot product. $\omega(e, 5)$ is the exponentially averaged weight vector to aggregate the predictions of the last 5 epochs to achieve more robust label update. The dynamic weighting factor, $\eta$, is used to adjust the degree of label update. As Figure 2 shows, $\eta$ linearly increases from 0 to 1 with $E_\eta$ epochs, gradually weakening the influence of the original labels. Besides, in the initial training stage, the model tends to improve independent predictions. As the accuracy of independent predictions increases, the model assigns a greater weight to the contextual predictions. We present the hyperparameter analysis experiment for $E_\eta$ in Appendix C.

## 4 EXPERIMENT

### 4.1 EXPERIMENTAL SETUP

**Datasets.** In this work, we use two public and one private BST data to measure the performance of models. More detailed descriptions can be found in Appendix D.

- **fNIRS.** The Tufts fNIRS to Mental Workload (**Tufts fNIRS2MW** (Huang et al., 2021)) data contains brain activity recordings and other data from adult humans performing controlled cognitive workload tasks. They label each part of the experiment with one of four possible levels of $n$-back working memory intensity. Following Huang et al. (2021), we classify 0-back and 2-back tasks.

- **Sleep.** The **SleepEDF** (Kemp et al., 2000) data contains PolySomnoGraphic sleep records for 197 subjects over a whole night, including EEG, EOG, chin EMG, and event markers, as well as some respiration and temperature data. In our work, following Kemp et al. (2000), we use the EEG Fpz-Cz channel and EOG horizontal channel.

- **SEEG.** The private SEEG data records brain signals indicative of suspected pathological tissue within the brains of seizure patients. Different neurosurgeons annotate the seizure waveforms within the brain signals for classification. In our work, we uniformly downsample the data to 250Hz and identify seizures for each single channel.

**Label disturbance.** We introduce a novel disturbance method to the original labels of the public data to simulate scenarios where labels are inconsistent. Specifically, we first look for the boundary points between different classes in a complete long time sequence. Then, we randomly determine with a 0.5 probability whether each boundary point should move forward or backward. Finally, we randomly select a new boundary point position from $r\%$ of the length of the class in the direction of the boundary movement. In this way, we can interfere with the boundary labels and simulate label inconsistency. Meanwhile, a larger value of $r\%$ indicates a higher degree of label inconsistency. In this work, we conduct experiments with $r$ values of 0, 20, and 40 for fNIRS and Sleep data.

**Baselines.**    We compare *Con4m* with state-of-art models from various domains, including one
time series classification (TSC) model with noisy labels **SREA** (Castellani et al., 2021), three image
classification models with noisy labels: **SIGUA** (Han et al., 2020), **UNICON** (Karim et al., 2022)
and **Sel-CL** (Li et al., 2022), one supervised TSC model **MiniRocket** (Dempster et al., 2021), one
time series backbone model **TimesNet** (Wu et al., 2023), and one time series forecasting model
**PatchTST** (Nie et al., 2023). See more detailed descriptions of the baselines in Appendix E.

Table 1: Overview of BST data used in this work.

| Data | Sample Frequency | # of Features | # of Classes | Subjects | Groups | Cross Validation | Total Intervals | Interval Length | Window Length | Slide Length | Total Segments |
|------|------|------|------|------|------|------|------|------|------|------|------|
| fNIRS | 5.2Hz | 8 | 2 | 68 | 4 | 12 | 4,080 | 38.46s | 4.81s | 0.96s | 146,880 |
| Sleep | 100Hz | 2 | 5 | 154 | 3 | 6 | 6,000 | 40s | 2.5s | 1.25s | 186,000 |
| SEEG | 250Hz | 1 | 2 | 8 | 4 | 3 | 8,000 | 16s | 1s | 0.5s | 248,000 |

**Implementation details.**    We use cross-validation (Kohavi, 1995) to evaluate the model's general-
ization ability by partitioning the subjects in the data into non-overlapping subsets for training and
testing. As shown in Table 1, for fNIRS and SEEG data, we divide the subjects into 4 groups and
follow the 2 training-1 validation-1 testing (2-1-1) setting to conduct experiments. We divide the
Sleep data into 3 groups and follow the 1-1-1 experimental setting. Therefore, we report the mean
values of 12 and 6 cross-validation results for fNIRS and Sleep data respectively. Notice that for
SEEG data, inconsistent labels already exist in the original data. To obtain a high-quality testing
group, we select one group for accurate labeling and use a majority voting procedure to determine
the boundaries. Then we leave the testing group aside and only change the validation group to report
the mean value of 3 experiments. We report the full experimental results in Appendix G.

**Evaluation metrics.**    We use Accuracy (Acc.) and Macro-$F_1$ ($F_1$) scores as our evaluation metrics
due to the balanced testing set. Macro-$F_1$ score is the average of the $F_1$ scores across all classes.

## 4.2    Label Disturbance Experiment

The average results over all cross-validation experiments of different methods are presented in Ta-
ble 2. Overall, *Con4m* outperforms almost all baselines across all data and all disturbance ratios.

Table 2: Comparison with state-of-the-art methods in the testing Accuracy (%) and $F_1$ score (%) on
three BST data. The **best results** are in bold and we underline the second best results.

| Model | | | | | | | | | | | | | | | | |
|------|------|------|------|------|------|------|------|------|------|------|------|------|------|------|------|------|
| | Noisy Label Learning | | | | | | Time Series Classfication | | | | | | Both | | | |
| | SIGUA | | UNICON | | Sel-CL | | MiniRocket | | TimesNet | | PatchTST | | SREA | | *Con4m* | |
| **Dataset** $r$% | Acc. | $F_1$ | Acc. | $F_1$ | Acc. | $F_1$ | Acc. | $F_1$ | Acc. | $F_1$ | Acc. | $F_1$ | Acc. | $F_1$ | Acc. | $F_1$ |
| fNIRS   0 | 64.58 | 67.37 | 63.21 | 61.15 | 63.92 | 63.86 | 60.89 | 61.28 | 65.17 | 67.47 | 52.87 | 51.79 | 65.18 | 70.10 | **67.91** | **71.28** |
| fNIRS   20 | 63.45 | 65.24 | 62.33 | 60.45 | 61.85 | 62.45 | 59.74 | 60.41 | 63.48 | 65.39 | 52.42 | 55.38 | 63.99 | 69.65 | **66.78** | **71.27** |
| fNIRS   40 | 60.55 | 63.47 | 60.63 | 57.35 | 61.21 | 61.75 | 57.56 | 57.87 | 61.76 | 63.45 | 51.94 | 52.67 | **63.75** | 69.40 | 63.50 | **70.04** |
| Sleep   0 | 54.47 | 54.28 | 62.71 | 62.26 | 63.43 | 63.48 | 62.36 | 62.00 | 59.87 | 59.50 | 58.72 | 58.40 | 49.73 | 48.81 | **67.93** | **68.02** |
| Sleep   20 | 53.50 | 53.07 | 62.59 | 61.63 | 63.19 | 63.45 | 62.17 | 61.75 | 59.17 | 57.72 | 56.69 | 56.16 | 49.43 | 48.80 | **66.61** | **66.31** |
| Sleep   40 | 52.16 | 51.32 | 60.65 | 58.34 | 61.85 | 61.72 | 59.19 | 58.38 | 56.68 | 55.73 | 54.21 | 53.05 | 48.22 | 45.72 | **65.34** | **64.31** |
| SEEG   - | 66.87 | 53.19 | 69.22 | 60.53 | 68.46 | 60.50 | 68.79 | 62.39 | 66.02 | 50.99 | 66.59 | 58.45 | 65.11 | 55.21 | **74.60** | **72.00** |

**Results of different methods.**    For fNIRS, *Con4m* achieves competitive performance compared to
SREA. SREA is particularly designed for time series data and could better identify the inconsistent
time segments in a self-supervised fashion. However, on Sleep and SEEG data that require a stronger
reliance on contextual information, SREA's performance is significantly lower than *Con4m*. More-
over, in the case of SEEG and Sleep data without disturbance, *Con4m* impressively improves **7.14**%
and **15.41**% compared with the best baseline in $F_1$ score. This results demonstrate the necessity of
considering contextual information when dealing with more complex independent segments.

**Results of different $r$%.**    Noisy label learning methods demonstrate close performance degradation
as $r$% increases from 0% to 20%. But with a higher ratio from 20% to 40%, SIGUA, UNICON,
Sel-CL and SREA show averaged 3.01%, 5.23%, 1.92% and 3.34% decrease in $F_1$ score across
fNIRS and Sleep data, while *Con4m* shows 2.37% degradation. For TSC models, there is a consis-
tent performance decline as $r$% rises. Non-deep learning-based MiniRocket shows a more robust

performance. The performance of PatchTST on fNIRS data exhibits significant instability, possibly due to its tendency to overfit inconsistent labels too quickly. The stable performance of *Con4m* indicates that our proposed training framework can effectively harmonize inconsistent labels.

**Results of random disturbance.** We also conduct experiments following the setting of random label disturbance, which is commonly employed in the noisy label learning works (Wei et al., 2021; Li et al., 2022; Huang et al., 2023) of the image classification domain. As shown in Figure 3(b), compared to our novel boundary disturbance, *Con4m* exhibits stronger robustness to random disturbance. Even with the 20% disturbance ratio, *Con4m* treats it as a form of data augmentation, resulting in improved performance. This indicates that overcoming more challenging boundary disturbance aligns better with the nature of time series data.

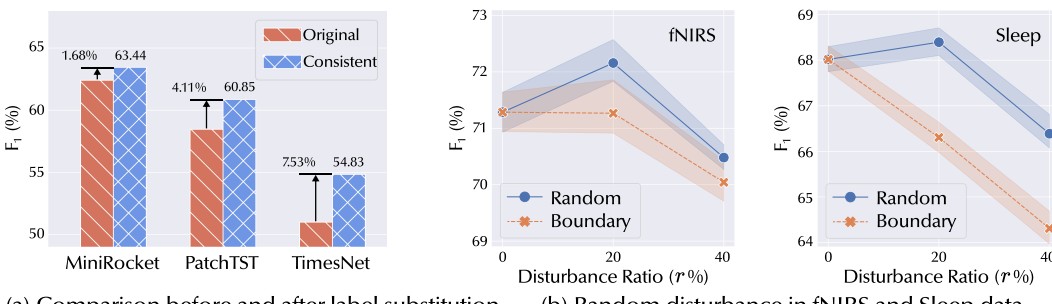

(a) Comparison before and after label substitution     (b) Random disturbance in fNIRS and Sleep data

Figure 3: Comparison results of label substitution and random disturbance experiments.

### 4.3 LABEL SUBSTITUTION EXPERIMENT

Since blurred boundaries are inherent to SEEG data and the majority voting procedure is costly, we limit this procedure to only one high-quality testing group in the label disturbance experiment. Besides, on the SEEG data, *Con4m* modifies approximately 10% of the training labels, which is a significant proportion. Therefore, it is necessary to further evaluate the effectiveness of our label harmonization process on SEEG data. Specifically, we train the TSC baselines based on the harmonized labels generated by *Con4m* and observe to what extent the baseline results are improved. As shown in Figure 3(a), PatchTST and TimesNet, employing deep learning architectures, are more susceptible to label inconsistency, so they obtain more significant performance improvement (4.11% and 7.53% in $F_1$ score). Unlike modified PatchTST that considers the classified segments in contexts, TimesNet only focuses on the independent segments, thus having a more dramatic improvement. In contrast, MiniRocket achieves only 1.68% increase. The reason may be that MiniRocket utilizes a simple random feature mapping approach without relying on specific patterns or correlations.

### 4.4 ABLATION EXPERIMENT

Table 3: Comparison with ablations in the testing Accuracy (%) and $F_1$ score (%) on two public data. The **best results** are in bold and we underline the second best results.

| Model | | | | | | | Preserve one | | | | | | Remove one | | | | | | | | | | |
|---|---|---|---|---|---|---|---|---|---|---|---|---|---|---|---|---|---|---|---|---|---|---|---|
| | | + Con-T | | + Coh-P | | + Cur-L | | - Con-T | | - Coh-P | | - Cur-L | | - Fit | | - $\eta$ | | *Con4m* | |
| **Dataset** | $r\%$ | Acc. | $F_1$ | Acc. | $F_1$ | Acc. | $F_1$ | Acc. | $F_1$ | Acc. | $F_1$ | Acc. | $F_1$ | Acc. | $F_1$ | Acc. | $F_1$ | Acc. | $F_1$ |
| Sleep | 20 | 65.97 | 65.05 | 65.76 | 65.10 | 65.31 | 64.76 | 65.73 | 65.53 | 65.84 | 65.07 | 65.85 | 65.43 | 66.06 | 65.28 | 62.02 | 59.97 | **66.61** | **66.31** |
| | 40 | 63.94 | 62.67 | 64.42 | 62.76 | 63.69 | 62.23 | 64.44 | 63.05 | 64.23 | 63.03 | 64.89 | 63.07 | 64.69 | 63.22 | 61.93 | 57.98 | **65.34** | **64.31** |
| SEEG | - | 71.68 | 67.85 | 71.69 | 69.04 | 71.32 | 67.22 | 73.85 | 70.59 | 72.41 | 68.26 | 74.17 | 71.18 | 73.47 | 70.63 | 70.70 | 66.04 | **74.60** | **72.00** |

We introduce two types of model variations. **(1) Preserve one module.** We preserve only the Con-Transformer (Con-T), Coherent Prediction (Coh-P), or Curriculum Learning (Cur-L) module separately. **(2) Remove one component.** In addition to removing the above three modules, we also remove the function fitting component (-Fit) and $\eta$ ($E_\eta = 0$) to verify the necessity of prediction behavior constraint and progressively updating labels.

As shown in Table 3, when keeping one module, +Coh-P achieves the best performance with averaged $2.78\%$ decrease in $F_1$ score, indicating that introducing the contextual class information are most effective for BST data. The utility of each module varies across datasets. For example, for Sleep data, the Con-T contributes more to performance improvement compared to the Cur-L module, while the opposite phenomenon is observed for SEEG data. As for removing one component, even when we only remove the Tanh function fitting, the $F_1$ score of *Con4m* significantly decreases $1.72\%$ on average. On the Sleep-$20\%$ and SEEG data, the drop caused by -Fit is more significant than that caused by some other modules. Moreover, the model variation -$\eta$ achieves the worst results ($9.23\%$ $F_1$ drop), aligning with our motivation. Specifically, during early training stages, the model tends to learn the consistent parts of the original labels. Premature use of unreliable predicted labels as subsequent training supervision signals leads to model poisoning and error accumulation.

## 4.5 CASE STUDY

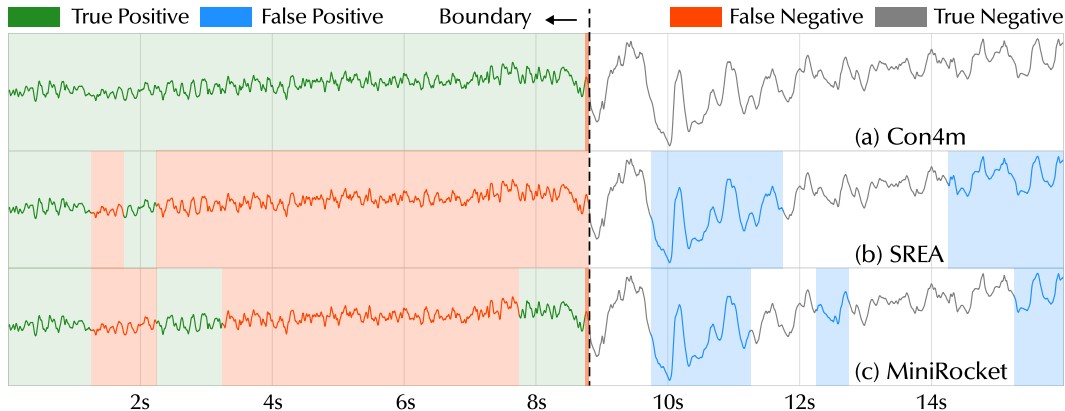

Figure 4: Case study for a continuous time interval in SEEG testing set.

We present a case study to provide a specific example that illustrates how *Con4m* works for BST data in Figure 4. We show a comparative visualization result of *Con4m*, SREA and MiniRocket for the predictions in a continuous time interval in SEEG testing set. In SEEG data, we assign the label of normal segments as $0$ and that of seizures as $1$. As the figure shows, *Con4m* demonstrates a more coherent narrative by constraining the prediction behavior to align with the contextual information of data. In contrast, MiniRocket and SREA exhibit noticeably interrupted and inconsistent predictions. What is even more impressive is that the model accurately identifies consistent boundaries within the time intervals spanning across two different states. This verifies that the harmonized labels capture the boundaries between distinct classes more precisely. Refer to Appendix H for more cases.

## 5 CONCLUSION AND DISCUSSION

In this work, we introduce the conception of Blurred-Segmented Time Series (BST) data and pose its unique challenges which have been overlooked by mainstream time series classification (TSC) models. Through theoretical analysis, we have obtained the conclusion that valuable contextual information enhances the predictive ability of the model. By introducing a novel method, *Con4m*, we incorporate effective contextual information at both the data and class levels to enhance model's predictive ability. Extensive experiments not only validate the superior performance achieved by *Con4m* through the integration of valuable contextual information, but also highlight the effectiveness and necessity of the proposed consistent label training framework for modeling BST data. Our approach still has some limitations. We have solely focused on analyzing and designing end-to-end supervised models. Further exploration to self-supervised methods would be challenging yet intriguing. When faced with more diverse label behaviors, the function fitting module needs to engage in more selection and design of basis functions. Nevertheless, our work brings new insights to the classification-based fields. In particular, for the TSC domain, we re-emphasize the importance of the inherent temporal dependence of time segments, shedding light on the era of personalized services.

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

## A    DETAILS OF RELATED WORKS

**Time series classification (TSC).** TSC has become a popular field in various applications with the exponential growth of available time series data in recent years. In response, researchers have proposed numerous algorithms (Ismail Fawaz et al., 2019). High accuracy in TSC is achieved by classical algorithms such as Rocket and its variants (Dempster et al., 2020; 2021), which use random convolution kernels with relatively low computational cost, as well as ensemble methods like HIVE-COTE (Lines et al., 2018), which assign weights to individual classifiers.

Moreover, the flourishing non-linear modeling capacity of deep models has led to an increasing prevalence of TSC algorithms based on deep learning. Various techniques are utilized in TSC: RNN-based methods (Rajan & Thiagarajan, 2018; Dennis et al., 2019) capture temporal changes through state transitions; MLP-based methods (Garnot & Landrieu, 2020; Wu et al., 2022) encode temporal dependencies into parameters of the MLP layer; and the latest method TimesNet (Wu et al., 2023) converts one-dimensional time series into a two-dimensional space, achieving state-of-the-art performance on five mainstream tasks. Furthermore, Transformer-based models (Yang et al., 2021; Chowdhury et al., 2022) with attention mechanism have been widely used.

The foundation of our work lies in these researches, including the selection of the backbone and experimental setup. However, mainstream TSC models (Middlehurst et al., 2023; Foumani et al., 2023) are often designed for publicly available datasets (Bagnall et al., 2018; Dau et al., 2019) based on the *i.i.d.* samples, disregarding the inherent contextual dependencies between classified samples in BST data. Although some time series models (Shao et al., 2022; Nie et al., 2023) use patch-by-patch technique to include contextual information, they are partially context-aware since they only model the data dependencies between time points, ignoring the class dependencies of segments.

**Noisy label learning (NLL).** NLL is an important and challenging research topic in machine learning, as real-world data often rely on manual annotations prone to errors. Early works focus on statistical learning (Angluin & Laird, 1988; Lawrence & Schölkopf, 2001; Bartlett et al., 2006). Researches including Sukhbaatar et al. (2015) launch the era of noise-labeled representation learning.

The label noise transition matrix, which represents the transition probability from clean labels to noisy labels (Han et al., 2021), is an essential tool. Common techniques for loss correction include forward and backward correction (Patrini et al., 2017), while masking invalid class transitions with prior knowledge is also an important method (Han et al., 2018a). Adding an explicit or implicit regularization term in objective functions can reduce the model's sensitivity to noise, whereas re-weighting mislabeled data can reduce its impact on the objective (Azadi et al., 2016; You et al., 2020; Liu et al., 2022). Other methods involve training on small-loss instances and utilizing memorization effects. MentorNet (Jiang et al., 2018) pretrains a secondary network to choose clean instances for primary network training. Co-teaching (Han et al., 2018b) and Co-teaching+ (Yu et al., 2019), as sample selection methods, introduce two neural networks with differing learning capabilities to train simultaneously, which filter noise labels mutually. The utilization of contrastive learning has emerged as a promising approach for enhancing the robustness in the context of classification tasks of label correction methods (Li et al., 2022; Zheltonozhskii et al., 2022; Huang et al., 2023).

These works primarily focus on handling noisy labels. And ensuring overall label consistency by modifying certain labels is crucial for BST data. To the best of our knowledge, the only noisy label learning work in the time series field is SREA (Castellani et al., 2021), which trains a classifier and an autoencoder with a shared embedding representation, progressively self-relabeling mislabeled data samples in a self-supervised manner. However, SREA does not take into account the contextual dependencies of BST data, limiting its performance.

**Curriculum learning (CL).** Bengio et al. (2009) propose CL, which imitates human learning by starting with simple samples and progressing to complicated ones. Based on this notion, CL can denoise noisy data since learners are encouraged to train on easier data and spend less time on noisy samples (Gong et al., 2016; Wang et al., 2021). Current mainstream approaches include Self-paced Learning (Kumar et al., 2010), where students schedule their learning, Transfer Teacher (Weinshall et al., 2018), based on a predefined training scheduler; and RL Teacher (Graves et al., 2017; Matiisen et al., 2020), which incorporates student feedback into the framework. The utilization of CL proves to be particularly advantageous in situations involving changes in the training labels. Hence, this technique is utilized to enhance the modeling of BST data in a more stable manner.

## B    IMPLEMENTATION DETAILS OF PREDICTION BEHAVIOR CONSTRAINT

To fit the hyperbolic tangent function (Tanh), we use the mean squared error (MSE) loss function. In practice, we use the Adam optimizer with a learning rate of $0.1$ to optimize the trainable parameters. The maximum number of iterations is set to $100$, and the tolerance value for stopping the fitting process based on loss change is set to $1e-6$. Sequences belonging to one minibatch are parallelized to fit their respective Tanh functions. To adapt to the value range of the standard Tanh function, we rescale the sequential predictions to $[-1, 1]$ before fitting.

However, it can be difficult to achieve a good fit when fitting with the Tanh function. Specifically, random initialization may fail to fit the sequential values properly when a long time series undergoes a state transition near the boundary. For example, as Figure 5(a) shows, we fit a sequence in which only the last value is $1$. We set all default initial parameters as $1$ and fit it. It can be observed that the fitting function cannot properly fit the trend and will mislabel the last point.

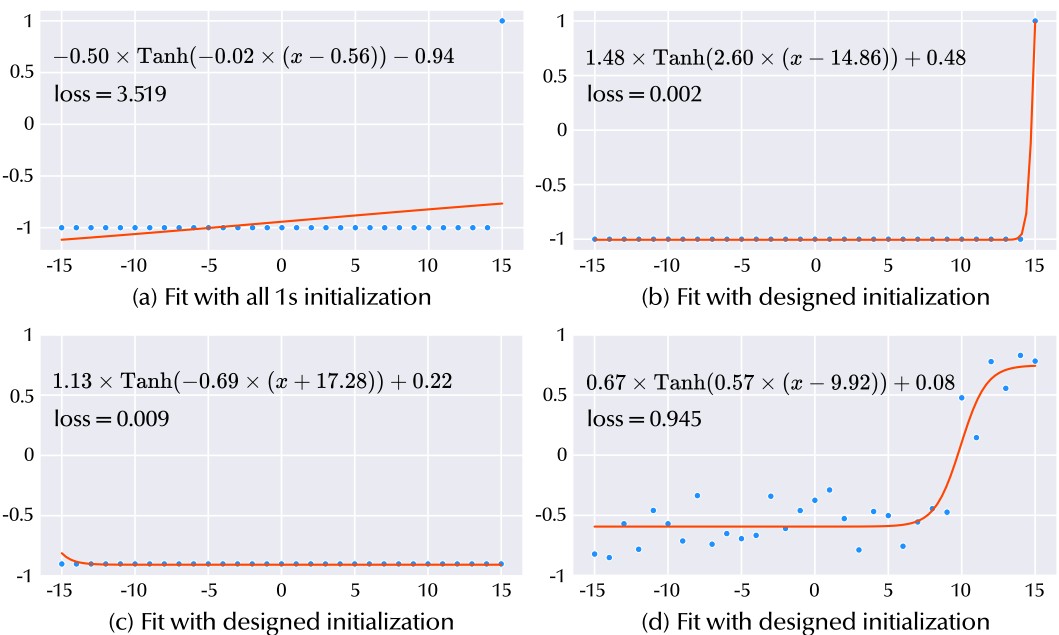

Figure 5:  Cases for Tanh fitting.

Appropriate parameter initialization is needed to avoid excessive bias. After careful observation, we find that parameter $k$ controls the slope at the transition part of Tanh, and parameter $b$ controls the abscissa at the transition point. In the process, all fitting values are assigned with uniform abscissa values. Therefore, we calculate the maximum difference between adjacent values and the corresponding position in the entire sequence. And these two values are assigned to parameters $k$ and $b$, respectively. This allows us to obtain suitable initial parameters and avoid getting trapped in local optima or saddle points during function fitting. Formally, given the $L$-length input sequence $\tilde{p}$, we initialize parameters $k$ and $b$ as follows:

$$di = \left[\tilde{p}_{i+1} - \tilde{p}_i\right]_{i \in \{1,\dots,L-1\}},\tag{19}$$

$$k, b = \max\left(\text{Abs}(di)\right), \arg\max\left(\text{Abs}(di)\right),\tag{20}$$

$$k = k \times \text{Sign}(di[b]),\tag{21}$$

$$b = -\left(b - \lfloor L/2 \rfloor + 0.5\right),\tag{22}$$

where $\text{Abs}(\cdot)$ and $\text{Sign}(\cdot)$ denote the absolute value function and sign function respectively. $di$ is the difference vector. After proper initialization, as Figure 5(b) shows, we can obtain more accurate fitting results to reduce the probability of mislabeling. We also show some other cases (Figure 5(c)(d)) for the fitting results to verify the effectiveness of the fitting process we propose.

## C  HYPERPARAMETER ANALYSIS

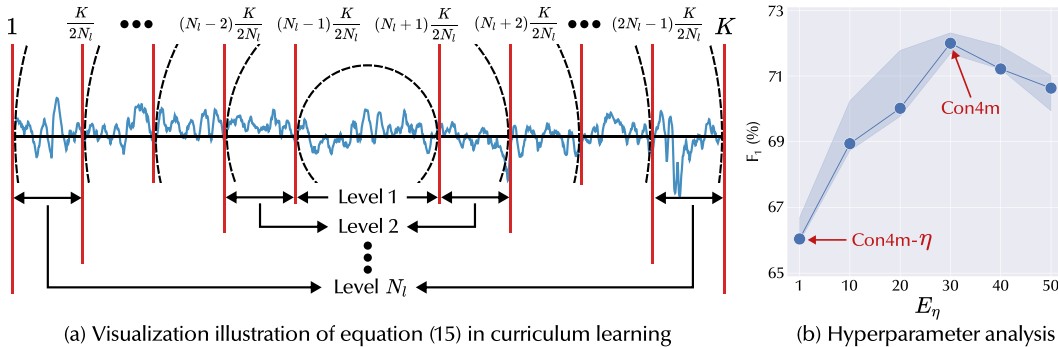

(a) Visualization illustration of equation (15) in curriculum learning    (b) Hyperparameter analysis

Figure 6: Visualization of data division in curriculum learning and hyperparameter analysis of $E_\eta$.

The dynamic weighting factor $\eta$ is introduced to progressively update the labels, preventing the model from overly relying on its own predicted labels too early. To validate the utility of $\eta$ and determine an appropriate linear growth epoch $E_\eta$, we conduct the hyperparameter search experiment on SEEG data. As shown in Figure 6(b), with smaller $E_\eta$ (corresponding to a higher growth rate), there is a significant improvement in model performance. This aligns with our motivation that during the early stage of model training, the primary objective is to better fit the original labels. At this stage, the model's own predictions are unreliable. If the predicted results are used as training labels too early in subsequent epochs, the model would be adversely affected by its own unreliability. On the other hand, excessively large $E_\eta$ leads to a slower rate of label updates, making it more challenging for the model to timely harmonize inconsistent labels. Nonetheless, considering the impact of variance, the model exhibits robustness to slightly larger $E_\eta$. In this work, we uniformly use $E_\eta = 30$ as the default value.

## D  DETAILS OF DATASETS

**fNIRS.** All signals are sampled at a frequency of $5.2$Hz. At each time step, they record 8 real-valued measurements, with each measurement corresponding to 2 concentration changes (oxyhemoglobin and deoxyhemoglobin), 2 types of optical data (intensity and phase), and 2 spatial positions on the forehead. Each measurement unit is a micromolar concentration change per liter of tissue (for oxy-/deoxyhemoglobin). They label each part of the active experiment with one of four possible levels of $n$-back working memory intensity (0-back, 1-back, 2-back, or 3-back). More specifically, in an $n$-back task, the subject receives 40 numbers in sequence. If a number matches the number $n$ steps back, the subject is required to respond accordingly. There are 16 rounds of tasks, with a 20-second break between each task. Following Huang et al. (2021), we only apply classification tasks for 0-back and 2-back tasks in our work. Therefore, we only extract sequences for 0-back and 2-back tasks and concatenate them in chronological order.

**Sleep.** The Sleep-EDF database records PolySomnoGraphic sleep data from 197 subjects, including EEG, EOG, chin EMG, and event markers. Some data also includes respiration and temperature-related signals. The database contains two studies: the Sleep Cassette study and the Sleep Telemetry study. The former records approximately 40 hours of sleep from two consecutive nights, while the latter records around 18 hours of sleep. Well-trained technicians manually score the corresponding sleep graphs according to the Rechtschaffen and Kales manual. The data is labeled in intervals of 30 seconds, with each interval being marked as one of the eight possible stages: W, R, 1, 2, 3, 4, M, or ?. In our work, we utilize only the data from the Sleep Cassette study, and retain only the signals from the EEG Fpz-Cz channel and EOG horizontal channel. The EEG and EOG signals were sampled at a frequency of $100$Hz. Following Kemp et al. (2000), we remove the labels for stages ? and M from the data, and merge stages 3 and 4, resulting in a 5-classification task.

**SEEG.** The private SEEG data records brain signals indicative of suspected pathological tissue within the brains of seizure patients. They are anonymously collected from a top hospital we coop-

erate with. For a patient suffering from epilepsy, 4 to 11 invasive electrodes with 52 to 153 channels are used for recording signals. In total, we have collected 847 hours of SEEG signals with a high frequency (1,000Hz or 2,000Hz) and a total capacity of 1.2TB. Professional neurosurgeons help us label the seizure segments for each channel. Before sampling for the database, we remove the bad channels marked by neurosurgeons. Then we uniformly downsample the data to 250Hz and use a low-pass filter to process the data with a cutoff frequency of 30Hz. Finally, we normalize and sample the intervals for each channel respectively.

## E    IMPLEMENTATION DETAILS OF BASELINES

- **SREA** (Castellani et al., 2021): This time series classification model with noisy labels jointly trains a classifier and an autoencoder with shared embedding representations. It gradually corrects the mislabelled data samples during training in a self-supervised fashion. We use the default model architecture from the source code provided by the author (https://github.com/Castel44/SREA).

- **SIGUA** (Han et al., 2020): This model adopts gradient descent on good data as usual, and learning-rate-reduced gradient ascent on bad data, thereby trying to reduce the effect of noisy labels. We modify the network for time series data based on the open source code provided by SREA, using the code from the author (https://github.com/bhanML/SIGUA).

- **UNICON** (Karim et al., 2022): UNICON introduces a Jensen-Shannon divergence-based uniform selection mechanism and uses contrastive learning to further combat the memorization of noisy labels. We modify the model for time series data according to the code provided by the author (https://github.com/nazmul-karim170/UNICON-Noisy-Label)

- **Sel-CL** (Li et al., 2022): Selective-Supervised Contrastive Learning (Sel-CL) is a latest baseline model in the field of computer vision. It selects confident pairs out of noisy ones for supervised contrastive learning (Sup-CL) without knowing noise rates. We modify the code for time series data, based on the source code provided by the author (https://github.com/ShikunLi/Sel-CL)

- **MiniRocket** (Dempster et al., 2021): Rocket (Dempster et al., 2020) achieves state-of-the-art accuracy for time series classification by transforming input time series using random convolutional kernels, and using the transformed features to train a linear classifier. MiniRocket is a variant of Rocket that improves processing time, while offering essentially the same accuracy. We use the code interface from the sktime package (https://github.com/sktime/sktime).

- **PatchTST** (Nie et al., 2023): This is a self-supervised representation learning framework for multivariate time series by segmenting time series into subseries level patches, which are served as input tokens to Transformer with channel-independence. We modify the code to achieve classification for each patch, based on the source code from the Time Series Library (TSlib) package (https://github.com/thuml/Time-Series-Library).

- **TimesNet** (Wu et al., 2023): This model focuses on temporal variation modeling. With Times-Block, it can discover the multi-periodicity adaptively and extract the complex temporal variations from transformed 2D tensors by a parameter-efficient inception block. We use the open source code from the TSlib package (https://github.com/thuml/Time-Series-Library).

## F    IMPLEMENTATION DETAILS OF *Con4m*

The non-linear encoder $g_{enc}$ used in *Con4m* is composed of three 1-D convolution layers. The number of kernels vary across different data and you can find corresponding parameters in the default config file of our source code. We construct the Con-Transformer based on the public codes implemented by HuggingFace[1]. We set $d = 128$ and the dimension of intermediate representations in FFN module as 256 for all experiments. The number of heads and dropout rate are set as 8 and 0.1 respectively. Since we observe that one-layer Con-Attention can fit the data well, we do not stack more layers to avoid overfitting. Note that *Con4m* consists of two Con-Transformers, we indeed use two Con-Attention layers. The model is optimized using Adam optimizer (Kingma & Ba, 2015) with a learning rate of $1e - 3$ and weight decay of $1e - 4$, and the batch size is set as 64. Also, we build our model using PyTorch 2.0.0 (Paszke et al., 2019) with CUDA 11.8. And the model is

---

[1]https://github.com/huggingface/transformers/blob/v4.25.1/src/transformers/models/bert/modeling_bert.py

trained on a workstation (Ubuntu system 20.04.5) with 2 CPUs (AMD EPYC 7H12 64-Core Processor) and 8 GPUs (NVIDIA GeForce RTX 3090). You can find more technical details in our source code, which has been attached in the supplementary materials.

# G    FULL RESULTS

The full results of the label disturbance experiment are listed in Table 4, 5 and 6. For fNIRS, we first divide the data into 4 groups by subjects and follow the 2 training-1 validation-1 testing (2-1-1) setting to conduct cross-validation experiments. Therefore, there are $C_4^2 \times C_2^1 = 12$ experiments in total. Similarly, we divide the Sleep data into 3 groups and follow the 1-1-1 experimental setting. Therefore, we carry out $C_3^1 \times C_2^1 = 6$ experiments. For SEEG data, we follow the same setting as fNIRS. Notice that we only select one group for accurate labeling to obtain a high-quality testing group, so we only have $C_3^2 = 3$ experiments. All the experimental results are listed in lexicographical order according to the group name composition. We also report the mean value and standard derivation of experiments for each data.

Table 4: Full results of the label disturbance experiment on **fNIRS** data. The **best results** are in bold and we underline the second best results.

| | | Noisy Label Learning | | | | | | Time Series Classfication | | | | | | Both | | | |
|---|---|---|---|---|---|---|---|---|---|---|---|---|---|---|---|---|---|
| | | SIGUA | | UNICON | | Sel-CL | | MiniRocket | | TimesNet | | PatchTST | | SREA | | *Con4m* | |
| $r\%$ | Exp | Acc. | $F_1$ | Acc. | $F_1$ | Acc. | $F_1$ | Acc. | $F_1$ | Acc. | $F_1$ | Acc. | $F_1$ | Acc. | $F_1$ | Acc. | $F_1$ |
| 0 | 1 | 62.01 | 64.75 | 62.18 | 63.85 | 63.06 | 63.95 | 60.89 | 61.37 | 61.14 | 60.73 | 51.61 | 51.07 | **65.06** | **69.13** | 64.61 | 68.55 |
| | 2 | 63.07 | 67.55 | 62.81 | 57.17 | 64.45 | 65.49 | 60.15 | 62.45 | 65.11 | 68.25 | 53.04 | 48.21 | 64.56 | 70.29 | **66.07** | **71.64** |
| | 3 | 62.93 | 65.56 | 60.16 | 61.71 | 63.67 | 61.44 | 60.81 | 60.96 | 63.99 | 66.38 | 51.70 | 54.23 | 65.66 | 67.14 | **66.20** | **70.51** |
| | 4 | 65.22 | 68.83 | 64.46 | 60.74 | 62.71 | 63.23 | 61.19 | 62.24 | 67.42 | 69.73 | 52.79 | 55.07 | 65.46 | 70.23 | **67.85** | **72.65** |
| | 5 | 63.28 | 67.96 | 61.70 | 54.87 | 61.46 | 63.21 | 60.46 | 61.35 | 63.07 | 66.46 | 52.99 | 57.11 | 64.48 | 70.13 | **66.54** | 68.55 |
| | 6 | 65.95 | 70.12 | 64.31 | 58.96 | 64.59 | 64.66 | 60.83 | 61.79 | 64.35 | 69.58 | 53.72 | 57.22 | 64.91 | 69.66 | **68.97** | **72.99** |
| | 7 | 61.14 | 64.03 | 60.74 | 62.54 | 61.85 | 60.13 | 60.38 | 60.11 | 61.72 | 62.64 | 52.44 | 50.46 | 64.89 | 70.55 | **68.34** | **70.63** |
| | 8 | 67.40 | 69.15 | 64.04 | 63.05 | 67.46 | 65.31 | 64.08 | 62.90 | 68.78 | 69.57 | 53.93 | 49.75 | 65.65 | 70.90 | **70.52** | **73.36** |
| | 9 | 65.76 | 68.41 | 63.12 | 66.77 | 61.52 | 63.45 | 58.60 | 58.78 | 64.31 | 67.23 | 51.97 | 48.86 | 64.98 | 70.29 | **69.37** | **72.60** |
| | 10 | 68.10 | 68.24 | 66.45 | 59.40 | 66.41 | 65.47 | 61.61 | 60.75 | 69.16 | 71.17 | 54.36 | 55.63 | 66.12 | 71.59 | 67.53 | 69.38 |
| | 11 | 64.53 | 65.95 | 63.24 | 57.84 | 64.84 | 65.24 | 59.12 | 60.71 | 64.18 | 66.64 | 51.65 | 48.51 | 63.71 | 69.17 | **65.75** | **69.42** |
| | 12 | 65.62 | 67.89 | 65.36 | 66.88 | 65.02 | 64.70 | 62.50 | 61.93 | 68.75 | 71.30 | 54.26 | 45.40 | 66.72 | 72.17 | **73.12** | **75.14** |
| | Avg | 64.58 | 67.37 | 63.21 | 61.15 | 63.92 | 63.86 | 60.89 | 61.28 | 65.17 | 67.47 | 52.87 | 51.79 | 65.18 | 70.10 | 67.91 | 71.28 |
| | Std | 2.14 | 1.87 | 1.85 | 3.72 | 1.90 | 1.69 | 1.44 | **1.13** | 2.75 | 3.23 | 1.02 | 3.91 | **0.80** | 1.28 | 2.36 | 2.11 |
| 20 | 1 | 62.66 | 61.42 | 62.55 | 64.38 | 60.10 | 60.91 | 58.75 | 59.22 | 62.40 | 62.74 | 51.37 | 49.81 | **64.89** | 68.32 | 64.02 | 69.48 |
| | 2 | 61.58 | 63.38 | 61.22 | 63.30 | 61.96 | 64.00 | 58.25 | 60.10 | 64.50 | 67.31 | 51.44 | 58.02 | 64.08 | 70.40 | **65.19** | **72.94** |
| | 3 | 63.82 | 64.27 | 60.13 | 51.67 | 57.58 | 57.25 | 60.26 | 59.52 | 59.67 | 60.19 | 51.79 | 54.99 | 61.94 | 70.16 | **67.50** | **72.06** |
| | 4 | 65.25 | 67.91 | 61.12 | 62.23 | 60.86 | 61.45 | 61.83 | 63.15 | 62.31 | 66.04 | 50.78 | 62.28 | 63.91 | 69.78 | **68.29** | **73.56** |
| | 5 | 62.54 | 66.07 | 63.31 | 56.26 | 61.93 | 64.22 | 60.97 | 62.04 | 64.07 | 67.66 | 55.10 | 55.29 | 61.60 | 68.45 | **67.82** | **71.41** |
| | 6 | 63.99 | 67.09 | 66.21 | 62.59 | 63.54 | 65.45 | 60.00 | 61.58 | 65.25 | 68.32 | 52.86 | 57.22 | 65.84 | 70.34 | **67.57** | **72.08** |
| | 7 | 60.54 | 60.97 | 59.68 | 49.65 | 60.96 | 59.18 | 59.15 | 59.15 | 59.49 | 59.02 | 52.09 | 53.13 | **63.57** | **67.69** | 59.02 | 62.68 |
| | 8 | 61.73 | 63.72 | 63.79 | 66.42 | 63.21 | 63.28 | 60.85 | 60.33 | 67.15 | 66.27 | 53.79 | 56.57 | 67.53 | 70.85 | **71.01** | **71.59** |
| | 9 | 64.50 | 67.11 | 58.42 | 62.28 | 61.56 | 61.54 | 58.58 | 59.41 | 61.38 | 66.73 | 52.95 | 52.13 | 62.00 | 68.15 | **68.97** | **71.84** |
| | 10 | 67.65 | 68.27 | 65.62 | 59.87 | 66.24 | 65.18 | 60.37 | 60.77 | 67.05 | 69.07 | 52.91 | 56.96 | 64.35 | 71.29 | 64.41 | **71.72** |
| | 11 | 62.84 | 64.83 | 63.91 | 63.95 | 61.18 | 63.32 | 57.25 | 58.87 | 63.64 | 65.17 | 50.23 | 49.62 | 63.90 | 69.48 | **67.61** | **72.08** |
| | 12 | 64.32 | 67.87 | 61.96 | 62.81 | 63.14 | 63.62 | 60.68 | 60.79 | 64.80 | 66.22 | 53.72 | 58.49 | 64.28 | 70.93 | **70.00** | **73.76** |
| | Avg | 63.45 | 65.24 | 62.33 | 60.45 | 61.85 | 62.45 | 59.74 | 60.41 | 63.48 | 65.39 | 52.42 | 55.38 | 63.99 | 69.65 | 66.78 | 71.27 |
| | Std | 1.90 | 2.53 | 2.36 | 5.22 | 2.12 | 2.46 | **1.34** | 1.32 | 2.52 | 3.15 | 1.40 | 3.71 | 1.68 | **1.22** | 3.22 | 2.92 |
| 40 | 1 | 58.40 | 60.63 | 59.09 | 52.63 | 61.46 | 61.98 | 57.46 | 57.21 | 59.92 | 62.93 | 52.20 | 51.39 | **63.60** | **69.37** | 60.14 | 65.90 |
| | 2 | 55.72 | 59.84 | 57.33 | 45.74 | 59.51 | 62.50 | 56.46 | 58.85 | 60.47 | 62.10 | 51.53 | 50.27 | 63.04 | 69.43 | **64.61** | **71.91** |
| | 3 | 61.09 | 65.00 | 58.31 | 54.70 | 60.57 | 59.58 | 57.75 | 58.30 | 60.77 | 60.06 | 51.29 | 44.09 | 62.83 | 69.12 | **66.02** | **71.05** |
| | 4 | 62.62 | 67.18 | 59.44 | 63.46 | 63.02 | 64.25 | 57.60 | 59.23 | 64.50 | 68.56 | 51.83 | 58.48 | 61.79 | 68.84 | **65.22** | **70.68** |
| | 5 | 61.58 | 64.60 | 64.05 | 63.00 | 57.99 | 58.31 | 56.78 | 57.05 | 60.37 | 59.96 | 51.18 | 54.44 | 62.43 | 68.49 | **64.34** | **71.55** |
| | 6 | 60.53 | 64.20 | 62.97 | 59.95 | 60.78 | 61.72 | 57.77 | 58.43 | 63.43 | 66.76 | 52.83 | 53.18 | 61.78 | 69.85 | **64.61** | **72.75** |
| | 7 | 59.17 | 61.58 | 59.64 | 52.33 | 61.44 | 60.33 | 56.30 | 56.25 | 59.08 | 60.06 | 51.11 | 49.93 | **63.75** | **69.19** | 60.12 | 66.69 |
| | 8 | 65.24 | 67.27 | 60.86 | 60.76 | 66.15 | 63.91 | 59.00 | 58.26 | 67.88 | 68.32 | 53.46 | 53.20 | 66.11 | 70.49 | 65.07 | 68.50 |
| | 9 | 57.08 | 61.80 | 62.72 | 65.20 | 61.13 | 62.82 | 56.73 | 56.95 | 61.19 | 63.86 | 49.79 | 61.90 | 61.59 | 68.42 | **65.23** | **69.88** |
| | 10 | 63.49 | 63.19 | 62.13 | 55.84 | 60.08 | 61.04 | 56.81 | 55.78 | 60.22 | 62.01 | 52.11 | 49.37 | **67.77** | **70.16** | 61.96 | 69.00 |
| | 11 | 59.14 | 62.12 | 61.70 | 57.92 | 59.85 | 61.68 | 57.92 | 58.34 | 60.20 | 62.78 | 52.94 | 57.77 | 62.81 | 68.36 | 59.84 | 70.59 |
| | 12 | 62.54 | 64.21 | 59.35 | 56.73 | 62.57 | 62.93 | 60.10 | 59.81 | 63.10 | 63.96 | 52.98 | 48.00 | **67.49** | **71.05** | 64.87 | 72.02 |
| | Avg | 60.55 | 63.47 | 60.63 | 57.35 | 61.21 | 61.75 | 57.56 | 57.87 | 61.76 | 63.45 | 51.94 | 52.67 | **63.75** | 69.40 | 63.50 | 70.04 |
| | Std | 2.76 | 2.38 | 2.08 | 5.57 | 2.06 | 1.73 | 1.10 | 1.22 | 2.52 | 3.03 | **1.03** | 4.95 | 2.18 | **0.85** | 2.30 | 2.14 |

Table 5: Full results of the label disturbance experiment on **Sleep** data. The **best results** are in bold and we underline the second best results.

| r% | Exp | Noisy Label Learning SIGUA Acc. | F1 | UNICON Acc. | F1 | Sel-CL Acc. | F1 | Time Series Classfication MiniRocket Acc. | F1 | TimesNet Acc. | F1 | PatchTST Acc. | F1 | Both SREA Acc. | F1 | Con4m Acc. | F1 |
|---|---|---|---|---|---|---|---|---|---|---|---|---|---|---|---|---|---|
| 0 | 1 | 54.74 | 54.79 | 63.40 | 62.41 | 63.86 | 63.49 | 62.80 | 62.16 | 59.92 | 58.73 | 58.95 | 58.42 | 49.76 | 48.95 | **69.31** | **68.80** |
| | 2 | 52.76 | 52.69 | 63.15 | 62.49 | 62.71 | 62.87 | 61.32 | 61.14 | 59.48 | 59.72 | 59.04 | 58.60 | 48.13 | 46.93 | **67.54** | **67.63** |
| | 3 | 56.24 | 56.19 | 63.40 | 62.62 | 65.73 | 65.47 | 63.49 | 62.74 | 61.47 | 60.76 | 60.21 | 59.44 | 50.32 | 49.38 | **69.14** | **69.29** |
| | 4 | 53.83 | 53.51 | 61.21 | 61.01 | 62.72 | 62.88 | 62.16 | 61.64 | 58.17 | 58.23 | 59.13 | 59.22 | 49.59 | 48.55 | **66.61** | **66.66** |
| | 5 | 54.82 | 54.36 | 63.19 | 62.89 | 63.62 | 63.82 | 62.30 | 62.25 | 61.14 | 60.80 | 57.83 | 57.45 | 49.79 | 48.82 | **66.55** | **66.61** |
| | 6 | 54.43 | 54.12 | 61.89 | 62.11 | 61.95 | 62.37 | 62.07 | 62.10 | 59.02 | 58.76 | 57.17 | 57.25 | 50.77 | 50.24 | **68.43** | **69.11** |
| | Avg | 54.47 | 54.28 | 62.71 | 62.26 | 63.43 | 63.48 | 62.36 | 62.00 | 59.87 | 59.50 | 58.72 | 58.40 | 49.73 | 48.81 | **67.93** | **68.02** |
| | Std | 1.15 | 1.19 | 0.93 | 0.66 | 1.32 | 1.10 | **0.73** | **0.55** | 1.26 | 1.10 | 1.07 | 0.90 | 0.90 | 1.09 | 1.22 | 1.22 |
| 20 | 1 | 54.24 | 53.73 | 63.38 | 62.75 | 64.13 | 64.41 | 62.30 | 61.86 | 59.58 | 58.07 | 57.14 | 56.82 | 50.00 | 49.80 | **67.57** | **67.07** |
| | 2 | 51.72 | 51.04 | 63.01 | 62.68 | 63.29 | 63.58 | 61.91 | 61.51 | 59.84 | 57.44 | 56.12 | 55.50 | 48.17 | 47.56 | **64.01** | **64.25** |
| | 3 | 54.68 | 54.51 | 62.44 | 61.44 | 64.29 | 64.58 | 62.95 | 62.35 | 59.51 | 56.10 | 57.53 | 57.03 | 50.63 | 49.30 | **68.76** | **68.50** |
| | 4 | 53.33 | 53.12 | 61.25 | 59.39 | 62.34 | 62.33 | 62.28 | 61.61 | 57.14 | 57.23 | 57.32 | 56.77 | 48.82 | 47.65 | **65.57** | **65.25** |
| | 5 | 53.20 | 52.83 | 62.72 | 61.92 | 63.15 | 63.28 | 61.90 | 61.75 | 60.00 | 58.99 | 55.18 | 54.78 | 48.48 | 48.18 | **66.26** | **65.90** |
| | 6 | 53.82 | 53.18 | 62.73 | 61.60 | 61.97 | 62.51 | 61.69 | 61.43 | 58.97 | 58.47 | 56.85 | 56.05 | 50.45 | 50.28 | **67.49** | **66.86** |
| | Avg | 53.50 | 53.07 | 62.59 | 61.63 | 63.19 | 63.45 | 62.17 | 61.75 | 59.17 | 57.72 | 56.69 | 56.16 | 49.43 | 48.80 | **66.61** | **66.31** |
| | Std | 1.03 | 1.16 | 0.73 | 1.22 | 0.93 | 0.94 | **0.45** | **0.33** | 1.06 | 1.02 | 0.89 | 0.89 | 1.06 | 1.16 | 1.69 | 1.50 |
| 40 | 1 | 53.08 | 52.10 | 60.95 | 58.17 | 61.83 | 61.54 | 59.57 | 58.62 | 57.61 | 57.20 | 56.78 | 55.98 | 48.99 | 47.23 | **66.79** | **65.38** |
| | 2 | 51.21 | 50.08 | 60.47 | 58.12 | 61.58 | 61.64 | 58.62 | 57.96 | 56.62 | 55.26 | 53.94 | 52.60 | 46.15 | 44.56 | **65.60** | **64.27** |
| | 3 | 54.12 | 53.85 | 61.02 | 59.63 | 63.70 | 63.27 | 60.03 | 59.18 | 55.81 | 55.30 | 53.72 | 52.12 | 48.97 | 47.98 | **66.31** | **65.36** |
| | 4 | 52.38 | 52.21 | 60.43 | 57.58 | 61.80 | 61.59 | 59.41 | 58.68 | 55.38 | 54.36 | 55.06 | 54.31 | 48.10 | 45.53 | **66.02** | **65.69** |
| | 5 | 50.99 | 49.48 | 59.88 | 57.16 | 61.44 | 61.28 | 58.45 | 57.47 | 57.43 | 56.80 | 52.47 | 50.80 | 48.39 | 44.03 | **63.54** | **61.82** |
| | 6 | 51.19 | 50.18 | 61.13 | 59.40 | 60.76 | 61.01 | 59.06 | 58.36 | 57.24 | 55.44 | 53.32 | 52.50 | 48.71 | 45.00 | **63.76** | **63.33** |
| | Avg | 52.16 | 51.32 | 60.65 | 58.34 | 61.85 | 61.72 | 59.19 | 58.38 | 56.68 | 55.73 | 54.21 | 53.05 | 48.22 | 45.72 | **65.34** | **64.31** |
| | Std | 1.26 | 1.68 | **0.48** | 0.99 | 0.98 | 0.80 | 0.60 | **0.60** | 0.92 | 1.06 | 1.51 | 1.82 | 1.07 | 1.56 | 1.36 | 1.51 |

Table 6: Full results of the label disturbance experiment on **SEEG** data. The **best results** are in bold and we underline the second best results.

| r% | Exp | Noisy Label Learning SIGUA Acc. | F1 | UNICON Acc. | F1 | Sel-CL Acc. | F1 | Time Series Classfication MiniRocket Acc. | F1 | TimesNet Acc. | F1 | PatchTST Acc. | F1 | Both SREA Acc. | F1 | Con4m Acc. | F1 |
|---|---|---|---|---|---|---|---|---|---|---|---|---|---|---|---|---|---|
| SEEG | 1 | 66.26 | 52.27 | 69.46 | 60.64 | 68.62 | 62.57 | 68.96 | 62.11 | 66.13 | 50.25 | 65.80 | 58.51 | 65.90 | 53.09 | **74.70** | **72.26** |
| | 2 | 67.93 | 55.21 | 71.09 | 64.43 | 64.73 | 51.65 | 69.02 | 61.12 | 65.70 | 50.55 | 68.12 | 60.60 | 64.29 | 57.57 | **75.23** | **73.21** |
| | 3 | 66.40 | 52.09 | 67.11 | 56.51 | 72.02 | 67.27 | 68.39 | 63.92 | 66.24 | 52.17 | 65.86 | 56.22 | 65.15 | 54.99 | **73.87** | **70.52** |
| | Avg | 66.87 | 53.19 | 69.22 | 60.53 | 68.46 | 60.50 | 68.79 | 62.39 | 66.02 | 50.99 | 66.59 | 58.45 | 65.11 | 55.21 | **74.60** | **72.00** |
| | Std | 0.93 | 1.75 | 2.00 | 3.96 | 3.65 | 8.01 | 0.35 | 1.42 | **0.28** | **1.03** | 1.32 | 2.19 | 0.81 | 2.25 | 0.69 | 1.36 |

## H CASE STUDY

As shown in Figure 7, we present four cases to compare and demonstrate the differences between our proposed *Con4m* and other baselines. The first two cases involve transitions from a seizure state of label 1 to a normal state of label 0. The third case consists of entirely normal segments, while the fourth case comprises entirely seizure segments. As illustrated in the figure, *Con4m* exhibits more coherent narratives by constraining the predictions to align with the contextual information of the data. Moreover, it demonstrates improved accuracy in identifying the boundaries of transition states. In contrast, MiniRocket and SREA exhibit fragmented and erroneous predictions along the time segments. This verifies that *Con4m* can achieve clearer recognition of boundaries, and it can also make better predictions on the continuous time segments belonging to the same class.

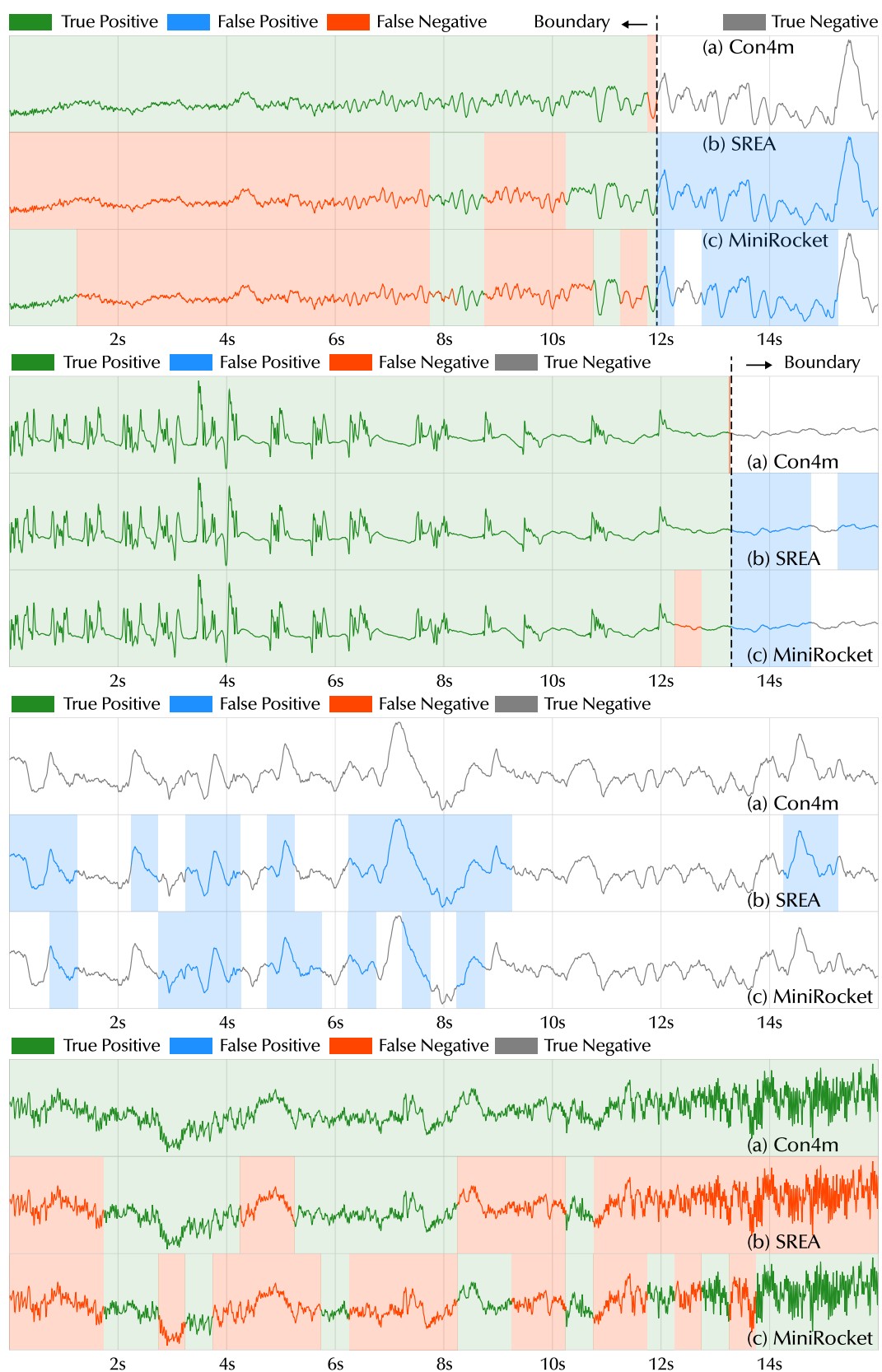

Figure 7: More cases for continuous time intervals in SEEG testing set.

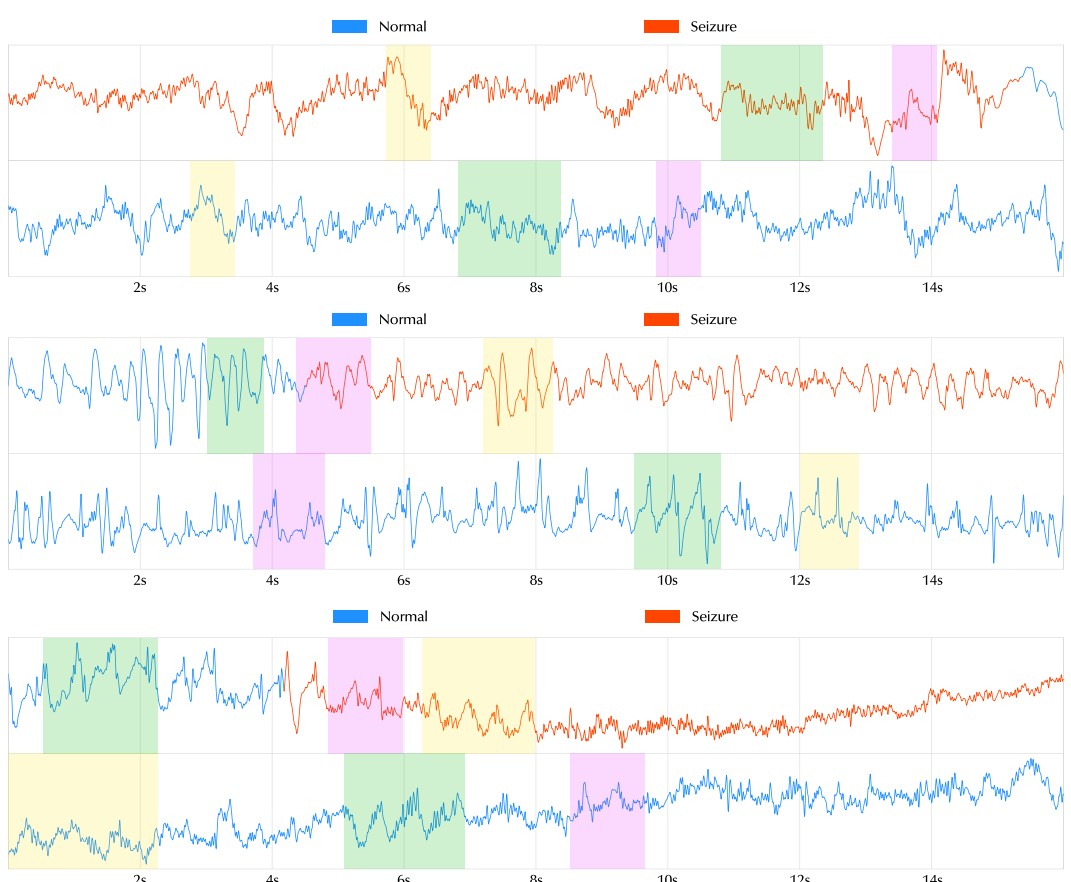

Figure 8: Comparison between boundary and central time intervals in SEEG data.

