# OpenReview forum: "Con4m: Unleashing the Power of Consistency and Context in Classification for Blurred-Segmented Time Series"
_ICLR.cc/2024/Conference — Submitted to ICLR 2024_

### Official Review · Reviewer_JdCs · 2023-10-15

**Soundness:** 2 fair
**Presentation:** 1 poor
**Contribution:** 2 fair
**Rating:** 3
**Confidence:** 4

**Summary:**

This paper proposes a blurred-segmented time series classification framework, Con4m, that forces label coherence and prediction behavior between two consecutive predictions. It also incorporates curriculum learning and gradual label change to cope with label inconsistency in transitions. Con4m shows its superiority in two public dataset and one private dataset with ablation studies for each component.

**Strengths:**

1. This paper covers a novel time-series data, blurred-segment time series.
2. Proposes practical framework for time series classification with noises.

**Weaknesses:**

1. Model degradation by label inconsistency in transition is not validated. The number of timestamp where transition occurs is very small comparing to the length of a whole time series. Does it really harm the model performance significantly? Plus, when annotating SleepEDF, multiple doctors are already recruited to make an agreement in their annotations, which can reduce inconsistency in state transition regions.

2. Methods seem to be a heuristic without enough justification and not novel. In neighbor class consistency discrimination, there could be so many ways to achieve it but there is no explanation on the design choice the authors made. Also, the theory does not support the reason why $\ell_2$ loss should be used.

3. Experiment setting is not convincing. The labels are disturbed synthetically and one of three datasets is a private dataset, which cannot be reproducible.

**Questions:**

1. What is the dimension of $x_t$? Is it different from $x_1,\ldots,x_L$? Is $x_t \in \mathbb{R}^{L \times d}$ where $d$ is the number of feature?

2. The function fitting incurs more computations in training loop. Can you elaborate on computational complexity?

3. At which layer $\ell_1$ and $\ell_2$ is applied?

---

> ### Author Response · Authors · 2023-11-14
> **Response to Questions**
>
> **Question 1:**
>
> - In the expressions related to mutual information and probability, $x_t$ represents the random variable of the input raw data. We have adopted the `\displaystyle \r` command as suggested in the ICLR template.
> - In Figure 1, $x \in \mathbb{R}^{L \times W \times d}=\{ x_1, \dots, x_L \}$, where $L$ represents the number of consecutive time segments, $W$ represents the window size (number of time points) in each segment, and $d$ represents the dimensionality of the input features.
> - All of $x_1,\dots,x_L$ are realizations of $x_t$.
>
> **Question 2:**
>
> - Function fitting does introduce additional computational overhead. However, **since we are fitting a Tanh function and the gradients do not backpropagate to the parameters of the encoder, such overhead is acceptable within the entire model**.
> - Let's assume the number of consecutive time segments in the input is $L$, the hidden representation dimension is $D$, and the number of classification categories is $C$. The Tanh function fitting involves $I$ internal iterative steps. The function fitting module fits the predictions for each category separately, where the prediction for each category in each time segment is a constant. Therefore, the time complexity of the function fitting module is $\mathcal{O}(ICL)$. If we only consider the final prediction linear layer in the encoder, which maps the hidden representation to the prediction logits, its complexity is $\mathcal{O}(DCL)$.
> - Hence, we suggest controlling $I<D$ in practical usage to effectively manage the time complexity of the function fitting module. In our experiments, we set the default parameters to $I=100$ and $D=128$.
>
> **Question 3:**
>
> As depicted on the left side of Figure 2, both $\ell_1$ and $\ell_2$ jointly guide the learning process of the entire encoder.

---

> ### Author Response · Authors · 2023-11-14
> **Response to Weaknesses**
>
> Dear Reviewer JdCs,
>
> Thanks for your questions and we carefully address your concerns as follows:
>
> **Weakness 1:**
>
> Solving the state transition problem holds significant practical significance.
>
> - **Harmonizing inconsistent boundary labels is an important means of reducing annotation costs.** As you mentioned, SleepEDF, being a widely used publicly available dataset, requires highly accurate labels. Achieving this level of precision necessitates the labor-intensive process of expert annotation, which is time-consuming. In the era of large-scale models, data and labels have become the core fuel for model performance. Unlike the field of NLP, medical annotations are precious and scarce. To reduce annotation costs and enable multiple institutions to collaborate on building the foundational model, harmonizing the annotations of different doctors is an essential step.
> - **Inconsistent boundary labels can also lead to a decrease in model performance.** Samples near the boundaries also appear throughout the entire time series. For instance, in SEEG data, there exists waveform overlap between the seizure waves and the interictal waves. As demonstrated in the case study presented in the paper, the repeated occurrence of samples near the boundaries confuses the model during training. Learning label consistency with contextual information can reduce the interference caused by inconsistent labels, allowing the model to converge more consistently on training samples.
> - **Accurately identifying boundaries is necessary for certain application scenarios.** For example, in the seizure detection task based on SEEG data, doctors need to determine the lesion areas by considering the temporal order of seizure waves generated by different channels (corresponding to different brain tissues). This is because the seizure waves will propagate from the lesion areas to surrounding brain tissues. The spread occurs rapidly, making it crucial to finely identify the boundaries of seizure onsets. Inconsistent labels prevent the model from providing precise and consistent predictions at the boundaries.
>
> **Weakness 2:**
>
> - The core insight of our theory lies in the selection of valuable contextual information. While people often focus solely on contextual information at the data level, **our theoretical framework allows for the incorporation of contextual information at the label level into the model's design**. In Section 3.2, lines 134-144, we mention that the theoretical variable $x_{\mathbb{A}\_t}$ can be replaced by $y_{\mathbb{A}\_t}$, benefiting from the analytical framework of information theory.
> - Based on theoretical analysis, the design in Section 3.2 revolves around incorporating $y_{\mathbb{A}_t}$ into the model's predictions. Introducing the neighbor class consistency discrimination task is the most intuitive approach. We explicitly discriminate whether contextual samples belong to the same category through supervised learning. Based on the output weights of the discriminator, **we directly incorporate the label information of contextual samples into the final predictions through a linear weighted sum**. This design is straightforward, concise, and effective.
>
> **Weakness 3:**
>
> - As you mentioned, publicly available datasets like SleepEDF require highly accurate labels. The absence of BST datasets is due to previous works ignoring the issue of label inconsistency. It is precisely because of this absence that we propose a novel label disturbance method for time series data (lines 219-226). Our experiments (lines 266-272) demonstrate that the label disturbance method we propose is more essential and challenging for time series data. **We have also included the codes in the attachment on how to disturb the publicly available datasets, which can be used by others.**
> - To validate the model's performance on more realistic datasets with inconsistent labels, we conducted experiments on SEEG. Apart from the data of the subjects in the test group, the data of other subjects were not manually harmonized (lines 239-242). **These experimental results can more realistically reflect the impact of label inconsistency on the model.** Due to strict ethical review, SEEG is highly sensitive medical data that cannot be publicly released at the moment. However, we are actively working towards collaborations with hospitals and subjects, hoping to share this data in a safer and restricted manner.

---

> > ### Comment · Reviewer_JdCs · 2023-11-20
> > **Response to author's rebuttal**
> >
> > Thanks for your response. I have further question as follows.
> >
> > 1. How much more time does Con4m take for its training comparing to training an encoder without fitting tanh? Is it the difference negligible?
> >
> > 2. Samples near the boundaries do not always appear throughout the entire time series. Transitions occur rarely and the data point neighboring boundaries would be also distinct when comparing to in-segment data points. If there are some examples, can you please refer the section or line number? Or, is there any numerical result that the label transition cause the performance degradation?

---

> > > ### Author Response · Authors · 2023-11-21
> > > **Response to further questions**
> > >
> > > Thanks for your response and new comments.
> > >
> > > |             | fNIRS        | Sleep        | SEEG         |
> > > | ----------- | ------------ | ------------ | ------------ |
> > > | Con4m       | 11.45s       | 37.73s       | 24.75s       |
> > > | w/o fitting | 1.71s        | 3.92s        | 6.78s        |
> > > |             | $\times$6.70 | $\times$9.63 | $\times$3.65 |
> > >
> > > 1. We conduct a simple timing test, and the table above reports the average time taken per epoch for training. According to the table, we do find that the Tanh function fitting module introduces additional time overhead. For the Sleep data, which involves a 5-classification problem, the additional computational cost is higher. In comparison to the fNIRS data, SEEG also involves a binary classification problem, but due to the larger window length of input samples and the total number of training samples, the additional overhead introduced by the function fitting is relatively smaller. Therefore, on datasets with a larger number of classes, the function fitting module incurs greater computational costs, which indeed is a limitation of our model design. On larger-scale datasets, the extra computational overhead from function fitting is reduced.
> > >
> > > 2. We would like to make clarifications through the following three points:
> > > - There are numerous transitions in the BST data. For example, in the case of fNIRS data, participants undergo alternating tests of multiple tasks. For Sleep data, there are significant variations in sleep stages during nighttime. In the case of SEEG data, the recording duration for participants is quite long and includes a considerable number of seizure occurrences. Each state change is accompanied by the generation of transition data.
> > > - The real challenge lies in the sampling of training data. We need to sample class-balanced examples from the training data to train the model, which inevitably requires determining class boundaries. Particularly for data like SEEG, the durations of different classes are extremely imbalanced. Therefore, for such data, the number of boundary samples sampled depends on the classes with shorter durations. Additionally, in the experimental setup described in Table 2, we disturb the boundaries between classes and then sample balanced training dataset based on the disturbed labels. As the perturbation ratio increases, the model's performance consistently declines, indicating that introducing label inconsistency near the boundaries harms the model's performance.
> > > - We also select three sets of sampled time intervals from a subject in the testing set of SEEG and present them in Figure 8 in Appendix H. These intervals are chosen from the mispredicted samples of the Con4m. For each set of intervals, the upper intervals are boundary data, while the lower samples are central data from the normal class. We aim to demonstrate that for physiological signals like SEEG, highly diverse waveforms contain extremely complex information. Even in time intervals far from the boundaries, similar waveforms or trends can appear. This is also a major challenge in SEEG-based seizure detection tasks. However, although such visualization results are illustrative, they are subjective and provided for reference purposes only.

---

### Official Review · Reviewer_zaSc · 2023-10-29

**Soundness:** 1 poor
**Presentation:** 1 poor
**Contribution:** 2 fair
**Rating:** 5
**Confidence:** 3

**Summary:**

The paper focuses on a time series classification problem in a novel setting of "blurred segmented" time series where each time series is exhaustively segmented and each segment is labeled with one of the given states. The notion of "blur" stems from the blurry transition boundaries between the two consecutive states in a given time series. The ultimate goal (to my understanding) is to train a Time Series Classification (TSC) model which can automate the segmentation and labeling process on such time series.  To train such a TSC model, the training data is comprised of labeled BS time series where the labels of all segment are manually annotated by multiple domain experts. The key feature of the proposed solution  is a novel deep learning attention-layer based architecture which is capable of leveraging the contextual dependencies between adjacent segments of time series. The proposed approach is evaluated against multiple baselines on three real-world datasets  (two public and one private) from healthcare and neuro-science domain which also appear to be the key applications of such work. The evaluation results seem to indicate better performance of the proposed approach in comparison to the baselines.

**Strengths:**

1. The problem setting of time series classification on blurred segmented time series is quite relevant for many domains.
2. The proposed approach seems to be outperforming the state-of-the-art time series classification baselines on multiple real-world datasets.
3. Ablation studies seem to justify the value of various components of the approach.

**Weaknesses:**

1. The paper is not written well and a bit difficult to follow. To begin with, the term "blurred segmented time series" is not concretely defined throughout the manuscript. To the best of my knowledge, this term is not ubiquitous in ML community. Further, the introduction does not clearly define the problem formulation.  In particular, it is not clear whether the end result is to classify the individual segments or classify an entire time series which comprises of multiple segments. The problem formulation is not mathematically defined even in subsequent sections which keeps a reader busy guessing. Further, several terms like samples vs segments vs timestamps , state vs labels are confusingly used at several places which makes it super difficult to understand the exact problem formulation. It is also not clear whether a segment is of fixed length or varying length.

2. The motivation regarding too much noise in the labels in the segments due to label inconsistencies on boundary segments is also not super convincing. For instance, why can't one simply get rid of  such boundary segments and train the model only on cleaner samples?

3. The theoretical justification section also lacks rigor and not quite convincing. In particular, the authors use mutual information definitions to make arguments in support of choosing augmented features from a neighborhood segment window. However, those arguments are very superficial and lack rigor (see detailed comments below).

4. The description of proposed approach is also quite difficult to follow. Several key notations are not well defined (e.g. what are V_s and V_t) and I had to read the papers in related work (e.g. Xu et al. 2022) in quite detail from where the ideas are borrowed. Even then, certain components of the approach such as neighbor class consistency discrimination are yet not clear to me.

**Questions:**

Specific comments/questions:
1. Page 3, line 86-87: This statement doesn't sound quite valid to me. What does it mean to say that we need to increase p(x_{A_t}|x_t)? We aren't talking about one specific value of x_{A_t} here, it's a distribution, right?  And ultimately all the terms are being summed up over all possible values of x_{A_t}. Similar concern for KL divergence argument. Basically, the justification given  in the support of design of proposed approach is not convincing and needs more rigor.

2. Page 3, lines 87-88: What do you exactly mean by "easier to predict"? Do you mean adding small noise to the samples? Perhaps being more specific here along with some citations will help.

3. A mathematical problem statement is dearly missing in Section 3.

4. Section 3.1: What is meant by a "smoother representation"? Perhaps you meant to say that the representation function should be "temporally smooth" so that the neighboring segments get embedded close-by in the embedding space?

5. Section 3.3, Lines 235 - 238: What is the significance of every group? How are you exactly getting 12 and 6 cross-validation results?

6. In section 4.5 (case study), what is the length of each segment? Is it fixed to 2 s? If so, how come SREA and MiniRocket has sub-segments of labels of lengths<2s? Or are we not labeling the entire segment with the same label?

7. Section 4.2, In lines 266-272: Is the noise coming due to challenging boundary disturbances similar to random noise as introduced in this experiment?

---

> ### Author Response · Authors · 2023-11-16
> **Response to Weaknesses**
>
> Dear Reviewer zaSc,
>
> Thank you sincerely for your thorough and valuable comments. We have carefully addressed your concerns and confusion by making relevant clarifications and modifications in the following response. We will also update the corresponding parts in the original manuscript and submit a revised version shortly.
>
> **Weakness 1 & Question 3:**
>
> Apologies for any confusion caused. Allow us to provide the following clarifications:
>
> - Blurred-Segmented Time Series (BST) refers to a type of time series data that exhibits distinct characteristics different from traditional Time Series Classification (TSC) data. In practical applications. BST data often arises, and it is characterized by two main aspects—Blurred Transitions and Prolonged State Durations. We establish this definition in lines 23-29 within the main context.
>
> - We divide a time interval into several consecutive segments and mainly focus on classification for each individual segment. We appreciate your feedback and we've incorporated a formal problem definition at the beginning of Section 3 as follows:
>
>   >**Definition 1.** Given a time interval comprising of $T$ consecutive time points, denoted as $s=\\{ s_1,s_2,\dots,s_T \\}$, a sliding window length of $w$ and slide length $r$ is employed for segmentation. The time interval is partitioned into $L$ time segments, represented as $x=\\{ x_i=\\{ s_{(i-1) \times r + 1},\dots,s_{(i-1) \times r + w} \\} | i=1,\dots,L \\}$. The model is tasked with predicting labels for each time segment $x_i$.
>
> - In our paper, we aim to employ 'segment' and 'state' when referring to specific scenarios, and in the context of machine learning, we prefer the terms 'sample' and 'label'. According to the problem definition above, our model focuses on predicting labels for each time segment. Therefore, 'sample' is equivalent to 'segment'. Also, the terms 'state' and 'label' are synonymous in our context, with each state having a corresponding label. Thank you for your recommendation and we will standardize and streamline the use of these terms in our paper.
>
> - According to the problem definition, each time segment maintains a fixed length in the same dataset, as indicated in Table 1.
>
> **Weakness 2:**
>
> - The nature of BST data inherently lacks a clear definition of boundaries. As described in lines 30-37, due to the absence of standardized quantification criteria, different doctors exhibit experiential variations in annotating boundaries. Moreover, this variation itself is ambiguous and lacks a precise definition. Therefore, the removal of boundary samples is not feasible in practice.
> - Employing a fixed length to remove boundary data is a crude and uncontrollable approach. Likewise, eliminating uncertain boundary samples through expert consensus would be prohibitively costly and unsustainable.
> - Some scenarios require precise boundary labels. For instance, in the seizure detection task based on SEEG data, doctors need to consider the temporal sequence of seizure occurrences in different brain regions to devise subsequent surgical strategies. If the model does not account for boundary labels during the training phase, it will struggle to accurately identify the class of samples at the boundaries during the inference phase.

---

> > ### Comment · Reviewer_zaSc · 2023-11-22
> > **Follow-up questions to authors' responses**
> >
> > I would like to thank authors for their detailed clarifications. Also thanks for explicitly clarifying the problem statement.
> > Some follow-up comments/questions on your response to Weakness 2:
> > 1. I am not quite convinced as to why eliminating uncertain boundary samples through expert consensus would be costly and unsustainable? In fact, you do the same to generate ground truth on SEEG dataset.
> > 2. For SEEG data, you mention that the precise boundary labels are indeed important, however your results are evaluated using the expert-consensus heuristic, which is what could have been used as a layman's approach. Ideally, it would be more assertive if there are any unique findings of proposed algorithm (which an expert consensus baseline will fail to identify) that are validated by a domain expert.
> > A convincing argument on at least one of the two points will be desirable.

---

> > > ### Author Response · Authors · 2023-11-22
> > > **Response to Follow-up questions to authors' responses**
> > >
> > > Thanks for your further questions and comments.
> > >
> > > 1. Firstly, the annotation of SEEG data requires trained annotators or experienced neurosurgeons. Many medical datasets necessitate expertise and training for proper annotation. This differs from the NLP and CV domains, where a large number of annotated samples can be collected through crowdsourcing. Secondly, a single subject's SEEG recording spans several days or even weeks (with a total data size of 1.2TB in our experiments) and includes hundreds of channels. Moreover, a participant's complete SEEG data often encompasses dozens or even more than one hundred of seizure occurrences. An annotator would require significant effort and time to review the entire recording of a participant and annotate **each channel**, let alone the collaborative effort needed to establish clear boundaries, as demonstrated in our testing set for SEEG. Lastly, to cover a diverse range of patients and seizure types, gathering data from multiple subjects requires collaboration among various institutions. Consequently, adopting algorithmic approaches to address boundary inconsistencies is a more realistic and cost-effective solution, aiming to minimize communication costs and reduce the burden on medical professionals.
> > >
> > > 2. Thank you for your suggestions. The objective of our paper has always been to address a machine learning problem abstracted from real-world scenarios. The model's capabilities remain within the constraints of expert annotations. We propose Con4m not to surpass the abilities of experts but to employ more practical algorithmic means to harmonize expert annotations (rather than relying on expert groups) in order to obtain a more stable model performance. Expert group annotations are necessary for validating the model's performance; otherwise, validating on a dataset with inherently inconsistent annotations would not assess the model's capabilities. Furthermore, for the fNIRS and Sleep data, we conducted validation by disturbing the boundaries of their respective training data. The validation on these two data was based on precise labels from publicly available datasets. Particularly for the Sleep data, its labels were also obtained through harmonization among expert groups. Of course, we fully agree with the ideal situation you mentioned, where the model can bring new knowledge and insights to experts. However, before achieving this ideal goal, our first priority should be to enable the model to adequately capture existing knowledge, assisting humans in tackling arduous and complex tasks, right?

---

> > > > ### Author Response · Authors · 2023-11-22
> > > > **Response to Follow-up comments to author's responses to Weakness 3 & Question 1-2**
> > > >
> > > > 1. In simple terms, increasing predictive ability refers to reducing the uncertainty of the label variable $y_{t}$ given the input sample variable $x_{A_{t}}$. In other words, we aim to make the distribution $p(y_{t}|x_{A_{t}},x_{t})$ approach a one-hot encoding form for the true labels. Initially, our intention was to present the theorem in a more natural language format to enhance reader comprehension. We also considered using mathematical notation, but it is indeed challenging to express it concisely in a formula. Do you have any suggestions on how to approach this?
> > > >
> > > > 2. Due to the convexity of the KL divergence, it exhibits monotonicity. As a result, the KL divergence attains its maximum value within the bounded space of $p(y_{t}|x_{A_{t}},x_{t})$ distributions, with the distribution that maximizes the KL divergence located at the boundary of the feasible space, i.e., the one-hot encoding distribution. This is equivalent to maximizing the predictive ability for the label variable. Even in practical scenarios, it may not be possible to achieve a $p(y_{t}|x_{A_{t}},x_{t})$ distribution that exactly matches the one-hot encoding form. However, the monotonicity property of the KL divergence still holds, indicating the existence of distributions $p(y_{t}|x_{A_{t}},x_{t})$ that maximize the KL divergence, with a larger value as the distribution approaches the boundary of the one-hot encoding. There can be multiple choices for $x_{A_{t}}$ that yield the maximum KL divergence, but this does not affect the conclusion of the theorem.
> > > >
> > > > 3. Your understanding is correct. Greater predictive ability corresponds to reduced uncertainty of $y_t$, i.e., the distribution $p(y_{t}|x_{A_{t}},x_{t})$ approaching the one-hot encoding form. We will make this more explicit in the main text. Alternatively, you can also view this problem from the perspective of information entropy. For discrete random variables with finite domains, as the distribution approaches the boundary, the information entropy decreases, indicating reduced uncertainty.
> > > >
> > > > 4. To achieve the upper bound on information gain, both of these conditions are necessary and can be simultaneously satisfied. Among them, finding the context set $A_{t}^{\*}$ that maximizes the KL divergence is crucial for the first inequality to hold with equality, $\mathbb{I}(y_t;x_{A_{t}}|x_t) = \sum_{x_t} p(x_t) \sum_{x_{A_{t}}} p(x_{A_{t}}|x_t) D_{t}^{\*}$. Furthermore, the optimal strategy distribution $p(x_{A_{t}^{\*}}|x_t)=1$ and $p(x_{A_{t}}|x_t)=0$ (for $A_t \neq A_{t}^{\*}$) ensures that the second inequality holds with equality, $\sum_{x_t} p(x_t) \sum_{x_{A_{t}}} p(x_{A_{t}}|x_t) D_{t}^{\*} = \sum_{x_t} p(x_t) D_{t}^{\*}$.

---

> ### Author Response · Authors · 2023-11-16
> **Response to Weaknesses**
>
> **Weakness 3 & Question 1-2:**
>
> Thank you for your valuable inquiry. We apologize for the oversight in this section. We have reorganized Theorem 1 as follows:
>
> **Theorem 1.** *Introducing a contextual sample set that maximizes the predictive ability of labels yields the maximum information gain.*
>
> *Proof.* Expanding $\mathbb{I}(y_t; x_{A_t}|x_t)$, we have:
> $$
> \begin{align}
> \mathbb{I}(y_t; x_{A_t}|x_t) &= \sum_{x_t} p(x_t) \sum_{x_{A_t}} \sum_{y_t} p(y_t,x_{A_t}|x_t) \log{\frac{p(y_t,x_{A_t}|x_t)}{p(y_t|x_t) p(x_{A_t}|x_t)}} \\\\
> &= \sum_{x_t}p(x_t) \sum_{x_{A_t}} \sum_{y_t} p(y_t|x_t,x_{A_t}) p(x_{A_t}|x_t) \log{\frac{p(y_t|x_t,x_{A_t})}{p(y_t|x_t)}} \\\\
> &= \sum_{x_t} p(x_t) \sum_{x_{A_t}} p(x_{A_t}|x_t) D_{KL} (p(y_t|x_t,x_{A_t}) \Vert p(y_t|x_t)).
> \end{align}
> $$
> Given a fixed classification sample $x_t$ and the inherent distribution $p(y_t|x_t)$ of the dataset, the KL divergence is a convex function that attains its minimum at $p(y_t|x_t,x_{A_{t}})=p(y_t|x_t)$. As $p(y_t|x_t,x_{A_{t}})$ approaches the boundary of the probability space, indicating a stronger predictive ability for $y_t$, the value of KL divergence increases. Due to the convexity of KL divergence, there exists a contextual sample set in the dataset that maximizes $D_{KL}(p(y_t|x_t,x_{A_{t}}) \\| p(y_t|x_t))$. We denote this sample set as $A_{t}^{\*}$ and the maximum KL divergence value as $D_{t}^{\*}$. Additionally, we note that $\sum_{x_{A_{t}}} p(x_{A_{t}}|x_t)=1$. Hence, we can obtain the upper bound for the information gain $\mathbb{I}(y_t;x_{A_{t}}|x_t) \le \sum_{x_t} p(x_t) \sum_{x_{A_{t}}} p(x_{A_{t}}|x_t) D_{t}^{\*} \le \sum_{x_t} p(x_t) D_{t}^{\*}$. To achieve this upper bound, the model needs to introduce a contextual sample set $A_{t}^{\*}$ for each sample that maximally enhances its label's predictive ability. Moreover, the model needs to reach an optimal selection strategy distribution $p(x_{A_{t}^{\*}}|x_t)=1, p(x_{A_{t}}|x_t)=0$ (for $A_t \neq A_{t}^{\*}$).
>
> ---
>
> According to Theorem 1, the model needs to find the optimal contextual sample set that enhances the predictive ability of each sample's label. In this paper, we utilize learnable weights to allow the model to adaptively select potential contextual sample sets. After aggregating contextual sample information, we train the model using sample labels. Through explicit supervised learning, the model can enhance its predictive ability for samples end-to-end while optimizing the selected contextual sample set. On the other hand, benefiting from an information-theoretic perspective, $x_{A_{t}}$ in Theorem 1 not only includes the raw data of contextual samples but also incorporates their label information ($y_{A_{t}}$). Therefore, we can introduce contextual information at both the data level (Section 3.1) and the label level (Section 3.2) to enhance the model's predictive ability.
>
> In the specific implementation, to balance computational costs, we limit the selection of contextual samples within a continuous and finite time range. At the data level, we introduce a learnable Gaussian kernel function to adaptively aggregate neighboring samples with a Gaussian prior, obtaining a smoother representation over time. At the label level, we explicitly train a neighbor class consistency discriminator. We aggregate the predicted results of contextual samples using the output probabilities of this discriminator as weights.

---

> > ### Comment · Reviewer_zaSc · 2023-11-22
> > **Follow-up comments to author's responses to Weakness 3 & Question 1-2.**
> >
> > I am afraid but the above explanation is still not satisfactory due to the following reasons:
> > 1. Firstly, the statement of theorem is not crisply defined. What do you exactly mean by "maximizes the predictive ability" ? Is there a way for you to mathematically state your theorem?
> > 2. You mention: "Due to the convexity of KL divergence, there exists a contextual sample set in the dataset that maximizes KL-divergence term". A convex function has a global minima, how can you say the same for maxima? For given x_t, you can have multiple choices of x_{A_t} that can maximize the KL divergence term. Basically it's not clear how are you getting any advantage out of the fact that the KL divergence (for given x_t) is convex in x_{A_t}.
> > 3. You mention: "As p(y_t|x_t, x_{A_t}) approaches the boundary of the probability space, indicating a stronger predictive ability for y_t, the value of KL divergence increases". The validity of this statement will be easier to assess once you clarify the notion of "stronger predictive ability". It sounds like you mean to say the distribution tends to attain extreme probability values for y_t, but this certainly needs to be clarified.
> > 4. You mention: "Moreover, the model needs to reach an optimal selection strategy distribution p(x_{A_t} | x_t) = 1 for x_{A_t} = x_{A_t}*". How does this necessarily align with the original statement in the theorem which only focuses on x_{A_t} that maximizes the "predictive ability of y". Are you trying to say that the two requirements are independent and can be simultaenously met? If so, please explain as that's not intuitive, at least to me.

---

> ### Author Response · Authors · 2023-11-16
> **Response to Weaknesses & Questions**
>
> **Weakness 4:**
>
> - As shown in the leftmost part of Figure 1, we implement the two-branch Con-Attention. $Q$, $K$ and $V$ vectors represent the query, key and value of the self-attention mechanism respectively. To distinguish between different computational branches, we use $s/S$ to represent the branch based on Gaussian prior, and $t/T$ to represent the branch based on vanilla self-attention. $\sigma$ is the learnable scale parameter of the Gaussian kernel function. $T^l$ and $S^l$ are the aggregation coefficients of the two branches in the $l$-th layer.
> - For the neighbor class consistency discrimination part, our framework allows for the incorporation of contextual information at the label level into the model's design. According to Theorem 1, we aim to identify a set of contextual samples that maximizes the model's predictive capability at the label level. Since directly optimizing the aggregation at the label level is challenging, we adopt the approach of aggregating predictions of samples belonging to the same class to enhance the model's predictive capability. This approach is inspired by the observation that for graph neural networks based on the homophily assumption, aggregating neighbor information belonging to the same class can improve predictive performance [1,2]. Therefore, we explicitly train a discriminator to determine whether two samples belong to the same class. The model selects a contextual sample set based on the discriminator's output probabilities and weights their predictions based on the probability of being classified as the same class.
>
> [1] McPherson, Miller, Lynn Smith-Lovin, and James M. Cook. "Birds of a feather: Homophily in social networks." *Annual review of sociology* 27.1 (2001): 415-444.
>
> [2] Zhu, Jiong, et al. "Beyond homophily in graph neural networks: Current limitations and effective designs." *Advances in neural information processing systems* 33 (2020): 7793-7804.
>
> **Question 4:**
>
> Thank you for seeking clarification. And your insight aligns exactly with what we mean. We will incorporate this clarification into our paper.
>
> **Question 5:**
>
> Each group is of the same significance in our analysis. To obtain the cross-validation results, we randomly partition the subjects within the dataset into non-overlapping subsets for both training and testing. For fNIRS, we first divide the data into 4 groups by subjects and follow the 2 training - 1 validation - 1 testing (2-1-1) setting to conduct cross-validation experiments. Therefore, there are $C^2_4 \times C^1_2=12$ experiments in total. Similarly, we divide the Sleep data into 3 groups and follow the 1-1-1 experimental setting. Therefore, we carry out $C^1_3 \times C^1_2=6$ experiments. The full contents are also elaborated in lines 679-685 of Appendix G.
>
> **Question 6:**
>
> - The reference coordinates "2s 4s ..." in Section 4.5 serve as a guide, while the actual window length (segment length) of SEEG data is 1s with a slide length of 0.5s, as detailed in Table 1.
>
> - During the labeling of time points from time segments, to reduce the error, we adopt a simple strategy for the time points at the intersection of two consecutive time segments. Specifically, the labels of the time points in the first half of the overlapping data align with the preceding time segment, while the latter half aligns with the subsequent time segment. For example, if the segment 0 consists of time points "1 2 3 4" with label "a", and the segment 1 comprises "3 4 5 6" with label "b". Then, time point 3 is assigned label "a", and time point 4 is assigned label "b".
>
>   | Label     | a    | a    | a    | b    | b    | b    |
>   | --------- | ---- | ---- | ---- | ---- | ---- | ---- |
>   | Segment 0 | 1    | 2    | 3    | 4    |      |      |
>   | Segment 1 |      |      | 3    | 4    | 5    | 6    |
>
> **Question 7:**
>
> Detailed implementation of the boundary disturbance is shown in lines 219-226 under the heading "Label disturbance". There exists a fundamental distinction between boundary disturbance and random noise. The boundary disturbance is employed to simulate scenarios where labels are inconsistent. The disturbance is introduced by disturbing the boundary points between different states. It is noted that only the results depicted in Figure 3(b) involve in random noise, which shows that boundary disturbance poses a more intrinsic and challenging problem compared to random noise for time series data. All of the other experiments are based on boundary disturbance.

---

### Official Review · Reviewer_5NtK · 2023-10-31

**Soundness:** 3 good
**Presentation:** 2 fair
**Contribution:** 2 fair
**Rating:** 6
**Confidence:** 3

**Summary:**

Blurred-segmented time series (BST) data has continuous states with inherently blurred transitions, leading to annotation noise. Existing time series classification methods do not consider label inconsistency and contextual dependencies between consecutive classified samples. To address these issues, the paper first theoretically identifies the value of contextual information, and then proposes a transformer-based model that incorporates contextual information from BST data. Moreover, the paper adopts curriculum learning techniques to train the model under annotation noise. Experiments show the proposed method achieves better classification accuracy than baseline methods on three datasets under different levels of label noise.

**Strengths:**

+ The problem setting is new and realistic. The paper consider time series classification on the new blurred-segmented time series data which has inherent contextual dependencies between classified samples. Without relying on the common i.i.d. assumption on samples, the proposed method boosts classification accuracy by explicitly exploiting the neighboring samples of a target sample.

+ The proposed method exploits both contextual information and noisy labels and can be applied to many realistic time series classification problems.

+ The experiments are well-designed and extensive. Results show that the proposed method outperforms baselines on the three datasets with different levels of label noise. Ablation studies show that each of the proposed components are effective in improving the time series classification accuracy.

**Weaknesses:**

- The clarity of the paper can be improved.
  Proposition 1 is a basic mutual information inequality. It is unclear how the mutual information $I(y_t;x_t,x_{\mathbb{A}_t})$ relates to the performance of a model.

- The proof of Theorem 1 mismatches with the claim.
  The proof only analyzes in what cases can $I(y_t;x_{\mathbb{A}_t}|x_t)$ be increased.
  How the predictive capability for the labels is defined? How do we know the contextual information enhances the predictive capability? And what is the connection between predictive capability and the mutual information gain?

- The motivation for the proposed method is not clear. For example, why using a Gaussian kernel function can better align with  $p(x_{\mathbb{A}_t}|x_t)$ and $p(y_t|x_t, x_{\mathbb{A}_t})$?

**Questions:**

1. In Proposition 1, how the mutual information $I(y_t;x_t,x_{\mathbb{A}_t})$ relates to the performance of a model.

2. In Theorem1, how the predictive capability for the labels is defined? How do we know the contextual information enhances the predictive capability? And what is the connection between predictive capability and the mutual information gain?

3. What is the computational complexity of the proposed method? Can the proposed method scale to longer time series?

4. Why using a Gaussian kernel function can better align with $p(x_{\mathbb{A}_t}|x_t)$ and $p(y_t|x_t,x_{\mathbb{A}_t})$?

---

> ### Author Response · Authors · 2023-11-17
> **Response to Weaknesses**
>
> Dear Reviewer 5NtK,
>
> Thank you for acknowledging our work. In the following response, we provide some clarifications and explanations:
>
> **Weakness 1 & Question 1:**
>
> - Specifically, mutual information measures the intrinsic properties of a dataset. For example, $\mathbb{I}(y_t;x_t)$ quantifies the correlation between sample and label distributions in the dataset from a probabilistic perspective. $\mathbb{I}(y_t;x_t,x_{A_{t}})$, on the other hand, describes the correlation between sample and label distributions when contextual samples are introduced. The theoretical framework is built upon the assumption that the model can perfectly capture these correlations (line 79). In other words, we seek the theoretical upper bound of the model's performance.
> - A larger value of mutual information indicates stronger correlation between variables. For instance, in the inequality $\mathbb{I}(y_t;x_t,x_{A_{t}}) \ge \mathbb{I}(y_t;x_t)$, introducing contextual samples $x_{A_{t}}$ enhances the correlation between the target sample $x_t$ and its label. In classification tasks, a higher correlation between input samples and labels implies that input samples are more easily distinguishable by their labels. In an ideal scenario, the model also possesses a higher upper bound on its performance to discriminate between samples.
>
> **Weakness 2 & Question 2:**
>
> Based on the previous answer, the increase in $I(y_t;x_{A_t}|x_t)$ determines the extent to which the upper bound of the model's performance improves. The purpose of Theorem 1 is to elucidate the specific contextual sample set that can maximize the correlation between the target sample and its label (i.e., $I(y_t;x_{A_t}|x_t)$). The following explanation is based on our latest version of Theorem 1 (Please see https://openreview.net/forum?id=EvBx5whpzJ&noteId=ZprzVTtrvl). In our proof, predictive capability is equivalent to the uncertainty of the conditional probability distribution of the label given the input sample. As the distribution $p(y_t|x_t,x_{A_{t}})$ approaches the boundary of the probability space, meaning the predicted probability for a certain class approaches 1 (and the probabilities for other classes approach 0), the uncertainty decreases. According to the proof, lower uncertainty corresponds to larger values of KL divergence. The upper bound of $I(y_t;x_{A_t}|x_t)$ is achieved when the KL divergence is maximized, indicating the lowest uncertainty. Therefore, we conclude that introducing contextual samples that maximally enhance predictive capability for the label results in the maximum information gain.
>
> **Weakness 3 & Question 4:**
>
> - According to Theorem 1, the model needs to find the optimal contextual sample set that enhances the predictive ability of each sample's label. In this paper, we utilize learnable weights to allow the model to adaptively select potential contextual sample sets. After aggregating contextual sample information, we train the model using sample labels. Through explicit supervised learning, the model can enhance its predictive ability for samples end-to-end while optimizing the selected contextual sample set.
> - The Gaussian kernel function serves as our prior knowledge for selecting the contextual sample set. BST data exhibits temporal persistence for each state (line 38). By paying closer attention to and aggregating neighboring samples, the model can acquire temporally smoother representations of time segments. Smoother representations lead to smoother predictive probabilities. This benefits not only the prediction of consecutive time segments belonging to the same category with the same label but also aligns with the gradual nature of class transitions (lines 110-114). According to the conclusions of the latest version of Theorem 1, the use of the Gaussian kernel function allows for a more targeted selection of the contextual sample set, thereby enhancing the model's predictive capability. We will revise these statements in the new version accordingly.

---

> > ### Comment · Reviewer_5NtK · 2023-11-22
> > **Follow-up comments to author's responses**
> >
> > I would like to thank authors for their clarifications. Here are some follow-up comments:
> >
> > - "the samples are more easily distinguishable by the labels" (line 79) does not make sense to me. In the data, is there any uncertainty for a sample to have a certain label?
> >
> > - I am not able to fully appreciate the novelty of Theorem 1. It seems that Theorem 1 does not bring too much insight.
> > The goal is to increase the predictive ability of a model using contextual samples. Theorem 1 just claims that the model needs to find the optimal contextual sample set that enhances its predictive ability (line 99).

---

> > > ### Author Response · Authors · 2023-11-23
> > > **Response to Follow-up comments to author's responses**
> > >
> > > Thanks for your response and follow-up comments.
> > >
> > > - Indeed, for a specific dataset, the training sample-label pairs are finite and deterministic. However, when discussing machine learning problems from a probabilistic perspective, we consider these sample-label pairs as (empirical) samples from the complete distribution of the dataset. In the theoretical analysis, we do not focus on a specific and deterministic dataset, but rather analyze the complete distribution of the dataset.
> > >
> > > - Theorem 1 represents a common practice in model design, where the use of context samples enhances the performance of the model (lines 68-71). In our theoretical analysis, Theorem 1 serves as a means to explain the existing practice. Additionally, it provides insights into optimizing the selection of contextual samples directly through an end-to-end approach based on the task objective, rather than relying on predefined methods.

---

> ### Author Response · Authors · 2023-11-17
> **Response to Questions**
>
> **Question 3:**
>
> The time complexity of Con4m can be divided into two main parts. The primary computational cost is on the same order as vanilla Transformer. Assuming the number of consecutive input time segments is $L$, the hidden representation dimension is $D$, the number of classes in the classification task is $C$, and the local iteration count of the function fitting module is $I$.
>
> - Con-Transformer: The time complexity of vanilla self-attention is $\mathcal{O}(LD^2+L^2D)$, and the time complexity of the Gaussian kernel branch is $\mathcal{O}(LD^2+L^2)$.
> - Coherent class prediction: The time complexity of Neighbor Class Consistency Discrimination is $\mathcal{O}(LDC)$, and the time complexity of the Tanh function fitting is $\mathcal{O}(ICL)$.
>
> Therefore, the computational cost and bottlenecks of Con4m are similar to vanilla Transformer. Large sequence lengths (numbers of time segments) and large hidden representation dimensions both increase the model's computational complexity. In contrast to the natural language processing field, where sequences are typically discrete, time series data consists of continuous numerical values. This allows for controlling the sequence length $L$ by adjusting the length of each time segment (patch). Thus, Con4m can be scaled to longer time series by appropriately tuning the patch length, the number of CNN layers, and the size of the convolutional kernels.

---

### Official Review · Reviewer_2FMY · 2023-11-04

**Soundness:** 3 good
**Presentation:** 3 good
**Contribution:** 3 good
**Rating:** 5
**Confidence:** 3

**Summary:**

In this paper, the authors study the blurred segmented time series (BST) data prediction problem. The authors theoretically clarify the connotation of valuable contextual information. Based on these insights, prior knowledge of BST data is incorporated at the data and class levels into the model design to capture effective contextual information. Moreover, the authors also propose a label consistency training framework to harmonize inconsistent labels. The authors have performed extensive experiments on real datasets to demonstrate the effectiveness of the proposed method in handling the time series classification task on BST data.

**Strengths:**

1.	The authors propose a new framework to handle the time series classification task on blurred segmented time series data.

2.	The authors provide some theoretical analysis about the connotation of the valuable contextual information.

3.	In the proposed framework, prior knowledge of the BST data at both the data and class levels are incorporated into the proposed model to capture the effective contextual information.

4.	The authors have performed extensive experiments on 3 real datasets to demonstrate the effectiveness of the proposed method.

**Weaknesses:**

1.	Some assumption of the proposed method seems a little strong. In Section 3.2, for the prediction behavior constraint, it is assumed that consecutive time segments span at most 2 classes within a suitably chosen time interval. The time interval may have a big impact on the model performance. However, it is not clear how to choose a suitable time interval for each dataset. The authors also need to perform experiments studying the impacts of the time interval on different datasets.

2.	The experimental analysis seems not consistent enough. In Figure 3(b), the analysis about random disturbance is studied on fNIRS and Sleep datasets. In Table 3, the ablation studies are performed on Sleep and SEEG datasets.

3.	The experimental analysis is not sufficient. Compared with existing methods, one advantage of the proposed method is to exploit the prior information at both the data and class levels. The authors are suggested to perform experiments studying the performance of the proposed method with only considering the prior information at data level and class level respectively.

**Questions:**

As discussed in Section 3.2, the time interval may have a big impact on the model performance. How to choose a suitable interval for each dataset?

---

> ### Author Response · Authors · 2023-11-17
> **Response to Weaknesses**
>
> Dear Reviewer 2FMY,
>
> Thank you for your comments and we address your concerns as follows:
>
> **Weakness 1:**
>
> Thank you for your valuable suggestion.
>
> - With domain knowledge, given the prior visual window size of manual annotation, we recommend selecting a slightly larger window size as the time interval length for two reasons:
>
>   - Typically, the annotation window does not contain data spanning more than two classes.
>   - Opting for a slightly larger window allows the model to leverage more contexts.
>
>   For instance, in the case of SEEG data, the doctors tend to use a 15-seconds visual window to identify the seizure waves. And based on empirical evidence, the duration of seizures is generally not shorter than 15 seconds. Hence, we chose a 16-second time interval to try to avoid the occurrence of multiple label transitions.
>
> - We are currently conducting experiments with time intervals scaled at 0.5, 1.5, and 2 times the initially selected length. Upon completion of the experiments, we will promptly provide the results and analysis.
>
>   To ensure the comparison to existing experimental parameters as fair as possible:
>
>   - We maintain identical experiment groups.
>
>   - We sample the same number of time intervals based on three different time interval lengths respectively. We proportionally scale both the window length and slide length, ensuring consistency in the number of time segments across all experiments.
>
>   - The hyperparameters of the main model and baselines remain consistent.
>
> **Weakness 2:**
>
> In Figure 3(b), we compare the performance of Con4m under the settings of random disturbance and boundary disturbance. The fNIRS and Sleep data are publicly available with accurate labels, allowing us to experiment with different disturbance strategies. However, for the SEEG data, we only have precise annotations for the test subjects, while the training data includes inconsistent annotations from different doctors (lines 238-242). Therefore, we cannot apply the random disturbance strategy to the SEEG training data. In the ablation experiments, we opted for the experimental settings with more significant boundary disturbance to clearly validate the effectiveness of each component of the model. Additionally, due to the constraints of the cognitive experimental setup, the transitions between different classes in the fNIRS data are more explicit, making them less susceptible to boundary disturbance. Hence, we selected Sleep-20%/40% and SEEG as the experimental groups for the ablation experiments.
>
> **Weakness 3:**
>
> We have conducted relevant experimental analyses. In the ablation experiments, - Con-T refers to replacing the Con-Transformer module with vanilla Transformer, thereby removing the prior information at the data level. - Coh-P indicates the simultaneous removal of the Neighbor Class Consistency Discrimination module and the function fitting module, meaning the elimination of the contextual information at the label level. - Fit represents the removal of only the function fitting module, implying the exclusion of prior trend information at the label level. Conversely, + Con-T and + Coh-P denote the retention of data-level and label-level contextual information, respectively.

---

> ### Author Response · Authors · 2023-11-20
> **Response to Weakness 1**
>
> We conduct the experiment about time interval length by selecting well-performing baselines and comparing them with Con4m. The results of the $F_1$ scores for each model are presented below:
>
> |            | fNIRS |       |       |       | Sleep |       |       |       | SEEG  |       |       |       |
> | ---------- | ----- | ----- | ----- | ----- | ----- | ----- | ----- | ----- | ----- | ----- | ----- | ----- |
> | $\times$   | 0.5   | 1.0   | 1.5   | 2.0   | 0.5   | 1.0   | 1.5   | 2.0   | 0.5   | 1.0   | 1.5   | 2.0   |
> | Sel-CL     | 61.07 | 63.86 | 63.90 | 64.50 | 59.06 | 63.48 | 66.34 | 68.24 | 61.18 | 60.50 | 59.18 | 58.13 |
> | MiniRocket | 60.32 | 61.28 | 60.37 | 61.49 | 57.17 | 62.00 | 65.30 | 67.57 | 60.96 | 62.39 | 61.19 | 61.90 |
> | SREA       | 67.55 | 70.10 | 69.46 | 70.91 | 47.13 | 48.81 | 49.98 | 52.30 | 56.14 | 55.21 | 52.09 | 50.28 |
> | Con4m      | 68.79 | 71.28 | 71.80 | 73.15 | 63.39 | 68.02 | 70.41 | 72.11 | 65.97 | 72.00 | 73.81 | 73.45 |
>
> Based on the experimental results, the following observations can be made:
>
> 1. **It is not recommended to choose time intervals shorter than the manually annotated visualization window.** For both fNIRS and Sleep data, all models perform the worst when the time interval was set at 0.5 times the window size, with a significant decrease in performance. This may be due to insufficient information captured by the smaller window size.
> 2. **Within the range from the visualization annotation window to the average durations of the shortest class, various time intervals can be chosen.** For both fNIRS and Sleep data, models show an improvement in performance as the time interval increases. This improvement is particularly significant in the Sleep data, which may be attributed to the labeling scheme assigning a label to each 30-second time window.
> 3. **It is recommended to consider both the characteristics of the data itself and the requirements of practical applications.** For the higher sampling rate and complexity of SEEG data, MiniRocket demonstrates more stable performance. However, Sel-CL and SREA show a decrease in performance as the window size increases. This can be attributed to the non-stationary nature of SEEG, which involves more diverse waveform variations with larger window sizes. Con4m, by considering contextual sample information, maintains relatively stable performance and even achieves higher performance. Additionally, to ensure comparability, we do not adjust the model parameters. When dealing with larger time intervals, it is necessary to consider increasing the model's parameter size appropriately.
> 4. In summary, Con4m consistently outperforms other models and exhibits a consistent performance improvement trend across all data. **This indicates the rationality and superiority of Con4m in addressing the classification problem in BST data.**

---

### Author Response · Authors · 2023-11-18

Dear reviewers,

We have revised and submitted a new version of the paper based on your feedback. We have highlighted the major changes in blue. Our revisions mainly involve the following:

- In response to Weakness 1 raised by Reviewer zaSc, we have reviewed and modified the usage of "sample"/"segment" and "state"/"label" throughout the paper.
- Addressing Weakness 1 mentioned by Reviewer 5NtK, we have added further discussion on the relevance of mutual information to model predictive performance in the proof of Proposition 1 (lines 77-81).
- In accordance with Weakness 3 highlighted by Reviewer zaSc, we have made corrections to Theorem 1 (lines 85-105).
- As suggested by Reviewer zaSc in Weakness 1, we have added the problem definition of the TSC task at the beginning of Section 3 (lines 112-115).
- Responding to Weakness 3 pointed out by Reviewer 5NtK, we have modified the association between the Gaussian kernel function and the theoretical part in Section 3.1 (lines 117-122).
- Following the feedback from Reviewer zaSc in Weakness 4, we have provided a description of intermediate variables in the Con-Attention module in Section 3.1 (lines 131-134).
- Based on Weakness 4 raised by Reviewer zaSc, we have adjusted the correlation between the Neighbor Class Consistency Discrimination module and the theoretical part in Section 3.2 (lines 148-155).
- Considering Weakness 1 from Reviewer 2FMY, we have added "the majority of" before the statement regarding time intervals spanning at most two classes in the Prediction Behavior Constraint section (line 163).

Thank you for your valuable feedback, and we believe that these revisions have significantly improved the clarity and quality of the paper. We sincerely anticipate receiving your feedback and will promptly address any new questions or comments.

---

### Meta-Review · Area_Chair_M8uY · 2023-12-08

**Metareview:**

In the original reviews, there were a lot of concerns regarding the motivation, theoretical studies, clarifications, and experimental results. During rebuttal, though some of the concerns are addressed, some concerns remain regarding the novelty of the theoretical results, arguments of the proposed method, etc. Therefore, based on the current shape, this work is not ready for publication.

**Justification For Why Not Higher Score:**

Some major concerns remain.

**Justification For Why Not Lower Score:**

N/A

---

### Decision · Program_Chairs · 2024-01-16

Reject